# A *Drosophila Su(H)* model of Adams-Oliver Syndrome reveals cofactor titration as a mechanism underlying developmental defects

Ellen K. Gagliani[1☯], Lisa M. Gutzwiller[2☯], Yi Kuang[3], Yoshinobu Odaka[4], Phillipp Hoffmeister[5], Stefanie Hauff[5], Aleksandra Turkiewicz[6], Emily Harding-Theobald[7], Patrick J. Dolph[7], Tilman Borggrefe[6], Franz Oswald[5], Brian Gebelein[2,8]*, Rhett A. Kovall[1]*

1 Department of Molecular Genetics, Biochemistry and Microbiology, University of Cincinnati College of Medicine, Cincinnati, Ohio, United States of America, 2 Division of Developmental Biology, Cincinnati Children's Hospital Medical Center, Cincinnati, Ohio, United States of America, 3 Graduate program in Molecular and Developmental Biology, Cincinnati Children's Hospital Research Foundation, Cincinnati, Ohio, United States of America, 4 Biology Department, University of Cincinnati Blue Ash College, Cincinnati, Ohio, United States of America, 5 University Medical Center Ulm, Center for Internal Medicine, Department of Internal Medicine, Ulm, Germany, 6 Institute of Biochemistry, University of Giessen, Giessen, Germany, 7 Department of Biology, Dartmouth College, Hanover, New Hampshire, United States of America, 8 Department of Pediatrics, University of Cincinnati College of Medicine, Cincinnati, Ohio, United States of America

☯ These authors contributed equally to this work.
* Brian.Gebelein@cchmc.org (BG); kovallra@ucmail.uc.edu (RAK)

**Data Availability Statement:** All relevant data are within the manuscript and its Supporting Information files.

## Abstract

Notch signaling is a conserved pathway that converts extracellular receptor-ligand interactions into changes in gene expression via a single transcription factor (CBF1/RBPJ in mammals; Su(H) in *Drosophila*). In humans, RBPJ variants have been linked to Adams-Oliver syndrome (AOS), a rare autosomal dominant disorder characterized by scalp, cranium, and limb defects. Here, we found that a previously described *Drosophila Su(H)* allele encodes a missense mutation that alters an analogous residue found in an AOS-associated RBPJ variant. Importantly, genetic studies support a model that heterozygous *Drosophila* with the AOS-like *Su(H)* allele behave in an opposing manner to heterozygous flies with a *Su(H)* null allele, due to a dominant activity of sequestering either the Notch co-activator or the antagonistic Hairless co-repressor. Consistent with this model, AOS-like Su(H) and Rbpj variants have decreased DNA binding activity compared to wild type proteins, but these variants do not significantly alter protein binding to the Notch co-activator or the fly and mammalian co-repressors, respectively. Taken together, these data suggest a cofactor sequestration mechanism underlies AOS phenotypes associated with RBPJ variants, whereby the AOS-associated *RBPJ* allele encodes a protein with compromised DNA binding activity that retains cofactor binding, resulting in Notch target gene dysregulation.

**Funding:** This work was supported by NSF/MCB-BSF grant #1715822 and #2114950 (B.G. and R.A. K.) and NIH grant CA178974 (R.A.K.); the Deutsche Forschungsgemeinschaft (German Research Foundation; SFB1074/A03, SFB1506/A05, OS 287/4-1 and GRK 2254/C4) and the "Deutsche Krebshilfe" (German Cancer Aid, #70114289) to F.O.; the Deutsche Forschungsgemeinschaft (DFG, German Research Foundation) [TRR81-A12 and BO1639/9-1]; State of Hesse (LOEWE iCANx) and the von Behring-Röntgen Stiftung (65-0004) to T.B. The funders had no role in study design, data collection and analysis, decision to publish, or preparation of the manuscript.

**Competing interests:** I have read the journal's policy and the authors of this manuscript have the following competing interests: R.A.K is on the scientific advisory board of Cellestia Biotech AG and has received research funding from Cellestia for projects unrelated to this manuscript. The remaining authors have declared that no competing interests exist.

## Author summary

Adams-Oliver Syndrome (AOS) is a rare disease defined by missing skin/skull tissue, limb malformations, and cardiovascular abnormalities. Human genetic studies have revealed that ~40% of AOS patients inherit dominant mutations within specific genes in the Notch signaling pathway. Notch signaling is a highly conserved cell-to-cell communication pathway found in all metazoans and plays crucial roles during embryogenesis and tissue homeostasis in organisms from *Drosophila* (fruit-flies) to mammals. The Notch receptor converts cell-to-cell interactions into a Notch signal that enters the nucleus and activates target genes by binding to a highly conserved transcription factor. Here, we took advantage of the unexpected finding that a previously described dominant allele in the *Drosophila* Notch pathway transcription factor contains a missense variant in an analogous residue found in a family with AOS. Using this novel animal model of AOS along with biochemical DNA binding, protein-protein interaction, and transcriptional reporter assays, we found that this transcription factor variant selectively compromises DNA binding but not binding to the Notch signal nor binding to other proteins in the Notch pathway. Taken together with prior human genetic studies, these data suggest AOS phenotypes associated with variants in the Notch pathway transcription factor are caused by a dominant mechanism that sequesters the Notch signal, leading to Notch target gene dysregulation.

## Introduction

Notch signaling is a highly conserved pathway that mediates cell-to-cell communication in all metazoans [1–3]. During embryogenesis, Notch plays a crucial role in vasculogenesis, hematopoiesis, neurogenesis, and cardiac development [4]. Additionally, Notch regulates tissue homeostasis, including epidermal differentiation and maintenance, lymphocyte differentiation, muscle and bone regeneration, and angiogenesis [4]. Intriguingly, Notch regulates these diverse processes using a common molecular cascade [5] (**Fig 1A**). Signaling is activated when Notch receptors on a cell interact with DSL (Delta, Serrate, Lag-2 for mammalian, *Drosophila*, and *C. elegans* orthologs, respectively) ligands on a neighboring cell. Mammals encode four Notch receptors (Notch1-4) and five DSL ligands (Dll1,3,4 and Jag1,2), whereas *Drosophila* has one Notch receptor and two ligands (Delta and Serrate). The Notch-DSL interaction triggers DSL endocytosis, resulting in force induced Notch cleavage [6], which is mediated by ADAM10 and the γ-secretase complex. Once cleaved, the Notch intracellular domain (NICD) is freed from the cell membrane, transits to the nucleus, and forms a complex with the transcription factor CSL (CBF1/RBPJ, Su(H) and Lag-1 for mammalian, *Drosophila*, and *C. elegans* orthologs, respectively) and the co-activator Mastermind (MAM) [7]. The NICD/CSL/MAM (NCM) complex binds to enhancer and promoter DNA elements to activate gene expression [8]. However, CSL can also function as a transcriptional repressor by forming complexes with co-repressor proteins, such as SHARP and Hairless in mammals and *Drosophila*, respectively [9–11]. Hence, Notch signal strength in a cell is largely defined by the balance of NCM activating complexes and CSL/co-repressor complexes that regulate target gene expression.

Genetic studies have shown that a subset of Notch-dependent processes are highly sensitive to gene dose. The term Notch derives its name from the original notched wing phenotype identified in *Drosophila*, resulting from Notch receptor haploinsufficiency. Heterozygous Notch phenotypes are also observable in the number and spacing of bristles on the fly notum. In knockout mice studies, Dll4 heterozygous-null alleles produce a severe phenotype with most pups dying *in utero* [12]. Moreover, human birth defects and developmental syndromes

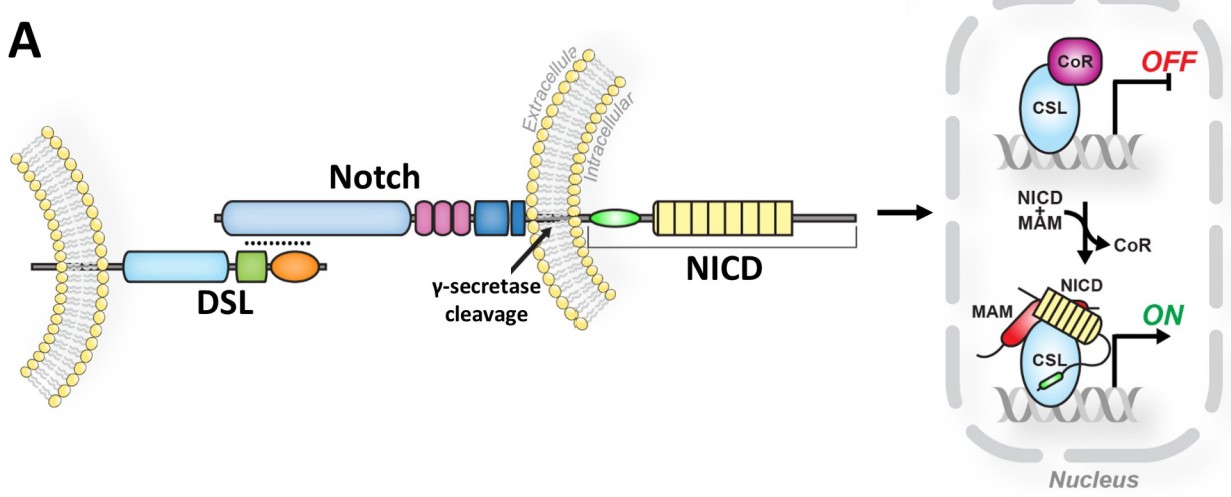

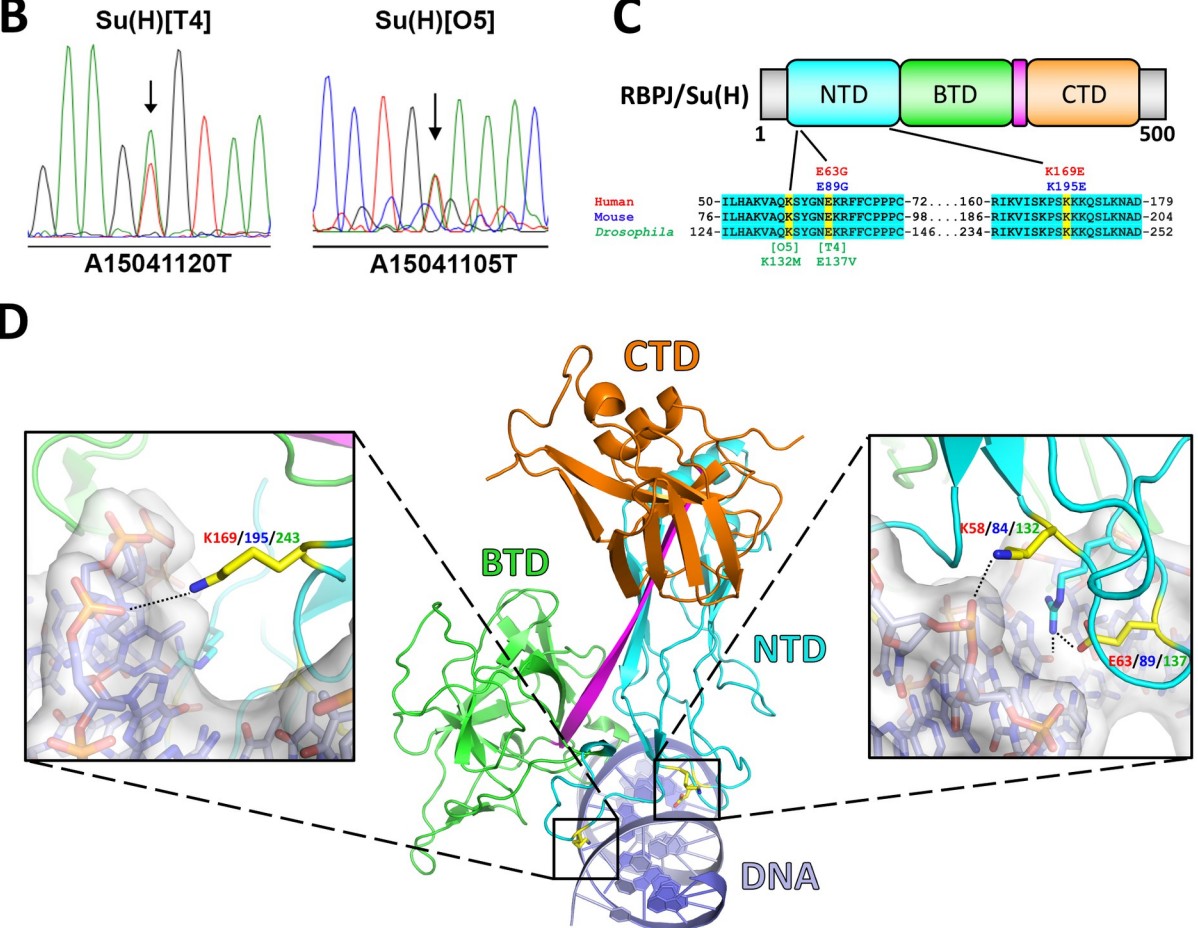

**Fig 1. Variants in the Notch pathway transcription factor associated with Adams Oliver Syndrome. A.** Overview of the Notch signaling pathway. Signal sending cells that express Notch ligands (DSL) interact with adjacent signal receiving cells that express Notch receptors. Upon receptor-ligand binding, the Notch receptor is cleaved by γ-secretase, resulting in the release of the soluble Notch Intracellular Domain (NICD). NICD binds to the CSL transcription factor and recruits the co-activator Mastermind (MAM) to activate transcription of Notch target genes. The CSL transcription factor can also engage co-repressors (CoR) to inhibit transcription. **B.** Sequence traces from PCR amplified genomic DNA

isolated from flies heterozygous for the *Su(H)*[T4] and *Su(H)*[O5] alleles reveal missense mutations (arrows) in each respective fly line. **C.** Top: Linear domain layout of CSL. NTD = N-terminal domain, BTD = beta-trefoil domain, CTD = C-terminal domain. Grey regions are disordered and poorly conserved between species. Bottom: Multiple sequence alignments of a portion of the human, mouse, and *Drosophila* NTD of CSL with residues that are 100% conserved highlighted in cyan. Yellow highlighting indicates the residues altered due to missense variants in two separate families with AOS (specific amino acid changes listed in red text) and the missense variants found in the *Su(H)* alleles (specific amino acid changes listed in green text). **D.** Middle: Ribbon diagram of RBPJ bound to DNA (PDBID 3IAG) [50], with the same domain coloring as described in C and the DNA shown in purple. Left: Close up view of the K169/195/243 residue in yellow, numbered according to human (red), mouse (blue), and *Drosophila* (green) respectively. Right: Close up view of the E63/89/137 and K58/84/132 residues in yellow, numbered according to the human (red), mouse (blue), and *Drosophila* proteins respectively. Black dashed lines indicate hydrogen bonding.

have been linked to haploinsufficiency for three out of the four Notch receptors (NOTCH1/2/3), two out of five ligands (DLL4 and JAG1), and the RBPJ transcription factor (human CSL) [13]. As an example, variants in the Notch pathway underlie Adams-Oliver Syndrome (AOS), a rare disease defined by scalp aplasia cutis congenita (missing skin and skull tissue) and limb malformations. Additionally, AOS patients can suffer cardiovascular abnormalities, such as cutis marmorata telangiectatica (dilated surface blood vessels), hypertension, and heart defects [14]. Genetic studies have revealed that ~40% of AOS patients inherit dominant mutations within the NOTCH1 receptor, the DLL4 ligand, or the RBPJ transcription factor. In contrast, haploinsufficiency mutations in the NOTCH2 receptor and JAG1 ligand are associated with Alagille Syndrome, a disease characterized by liver, eye, kidney, heart, skeleton, and vasculature defects [15]. Hence, genetic variants within the Notch signaling pathway can cause developmental defects in organisms from flies to humans.

In 2012, Hassad *et. al.* reported two autosomal dominant variants in RBPJ in separate families with AOS [14]. Interestingly, this study was the first to report germline variants in RBPJ that cause a disease in humans. These variants (E63G & K169E) mapped to the DNA binding region of RBPJ (**Fig 1C and 1D**), and the authors used cell extracts to show each variant impaired DNA binding [14]. Hence, these variants were classified as loss-of-function alleles. Here, we show that a previously identified allele in *Su(H)* (*Drosophila* CSL), *Su(H)*[T4], encodes an E137V mutation at the same highly conserved glutamic acid residue (human RBPJ E63G) as seen in AOS patients. However, prior genetic studies showed that *Su(H)*[T4] has conflicting "gain-of-function" activity as heterozygotes in *Drosophila* and "loss-of-function" activity in homozygous mutant clones [16,17]. To better explain the discrepancy between these conflicting biochemical and genetic results, we used a combination of quantitative *Drosophila* genetic studies, DNA binding assays, and protein-protein interaction assays. Our data show that in heterozygous flies the AOS-like *Su(H)* allele and a *Su(H)* null allele behave in a largely opposing manner, as the *Su(H)* AOS-like allele exacerbated both *Notch* and *Hairless* phenotypes, whereas the *Su(H)* null allele partially suppressed *Notch* and *Hairless* phenotypes. Moreover, we found that the *Su(H)* alleles and the mammalian *RBPJ* AOS variant alleles encode proteins defective in DNA binding. However, the AOS-like variants bound both the Notch activation complex and the Hairless/SHARP repression complexes with similar affinities as wild type Su(H)/RBPJ. Altogether, these data suggest a sequestration mechanism, in which RBPJ/Su(H) AOS variants efficiently bind the Notch pathway co-activator and co-repressor proteins, but their reduced DNA binding activity excludes these complexes from DNA and results in the mis-regulation of Notch target genes.

## Results

### Identification of a *Drosophila Su(H)* allele with an analogous amino acid change as a human *RBPJ* variant associated with AOS

Genetic studies in *Drosophila* have previously identified several *Su(H)* alleles that were described as having dominant gain-of-function phenotypes [17]. To better understand the

molecular function of these alleles, we sequenced the *Su(H)* coding regions of the *Su(H)^T4* and *Su(H)^O5* alleles and found that both contain a single missense mutation within the same highly conserved domain (i.e. the N-terminal domain, or NTD) of Su(H), which is involved in DNA binding (**Fig 1B**). *Su(H)^O5* encodes a lysine to a methionine mutation (K132M), whereas *Su (H)^T4* encodes a glutamic acid to valine mutation (E137V). Intriguingly, the E137 Su(H) residue corresponds to E63 in human RBPJ, which was found to be encoded by a missense variant (E63G) in a family with Adams Oliver Syndrome (AOS) [14] (**Fig 1C**). However, unlike the dominant gain-of-function activity ascribed to the fly *Su(H)^T4* allele in heterozygous animals, the RBPJ E63G variant was proposed to be a loss-of-function allele due to loss of DNA binding activity. Based on the crystal structures of RBPJ and Su(H) bound to DNA, both AOS variants identified by Hassed *et. al.* [14] (E63G & K169E) and both alleles identified by Fortini *et. al.* *[17]* (K132M and E137V) are residues involved in specific and non-specific interactions with DNA (**Fig 1D**). It should be mentioned that while E63 and E137 correspond to the same glutamate residue in human and *Drosophila* RBPJ/Su(H), K169 and K132 are not homologous. Nonetheless, the discrepancy between the predicted gain-of-function activity of the Su(H) E137V variant in *Drosophila* and the predicted loss-of-function activity of the analogous human E63G variant provided an opportunity to reexamine the mechanisms underlying how the homologous Su(H) and RBPJ variants impact Notch signaling.

## Su(H) and RBPJ AOS variants disrupt DNA binding

To elucidate how the RBPJ/Su(H) variants affect function, we first performed comparative DNA binding assays using EMSAs with a probe containing a RBPJ/Su(H) binding site and purified recombinant proteins corresponding to *Drosophila* Su(H) and mouse Rbpj (**Fig 2A and 2B**, note we will refer to the mouse Rbpj protein with only the first letter capitalized, whereas human RBPJ is in all capital letters). We expressed and purified WT and AOS variants of mouse Rbpj (53–474) and *Drosophila* Su(H) (98–523), which corresponds to the highly conserved structural core of CSL proteins [7]. We used differential scanning fluorimetry to determine if the variants affect folding and overall stability of Rbpj/Su(H). Consistent with these residues being surface exposed, none of the variants significantly affected folding/stability of Rbpj/Su(H) compared to WT proteins except for mouse Rbpj E89G (analogous to human RBPJ E63G), which had a modest effect on its melting temperature ($\Delta T_m$ = -3.1˚C, **S1 Fig**). EMSAs were performed with equimolar concentrations of WT Su(H) and the E137V [encoded by the *Su(H)^T4* allele] and K132M [encoded by the *Su(H)^O5* allele] variants, which revealed that both variants decreased, but did not abolish, DNA binding to a high affinity site (**Fig 2A**). Quantitation of the EMSA data revealed that Su(H) E137V consistently had a stronger impact on DNA binding than the Su(H) K132M variant. Similarly, purified Rbpj variants carrying analogous mouse amino acid variants (E89G and K195E) as those residues found in human associated AOS variants (E63G and K169E) also strongly weakened but did not abolish DNA binding (**Fig 2B**). In this case, however, K195E had an even greater impact on DNA binding than E89G (**Fig 2B**). These results are further supported by EMSAs performed with the full-length mammalian proteins produced by cell-free *in vitro* transcription/translation (**S2 Fig**).

Next, we used isothermal titration calorimetry (ITC) to quantitatively assess the changes in affinity and thermodynamics of these variant proteins binding to DNA (**Fig 2C and 2D**, **S1 Table**). Each binding experiment was performed by titrating DNA containing a single Rbpj/Su (H) binding site from a syringe into a calorimetric cell containing either purified Su(H) or Rbpj. WT Su(H) bound to DNA with a $K_d$ of 188 nM (**Fig 2C**, **S1 Table**); whereas Su(H) K132M bound to DNA with a 690 nM $K_d$, showing an ~3.5 fold reduction in binding, and Su (H) E137V bound to DNA with a $K_d$ of 842 nM displaying an ~4.5 fold reduction in binding

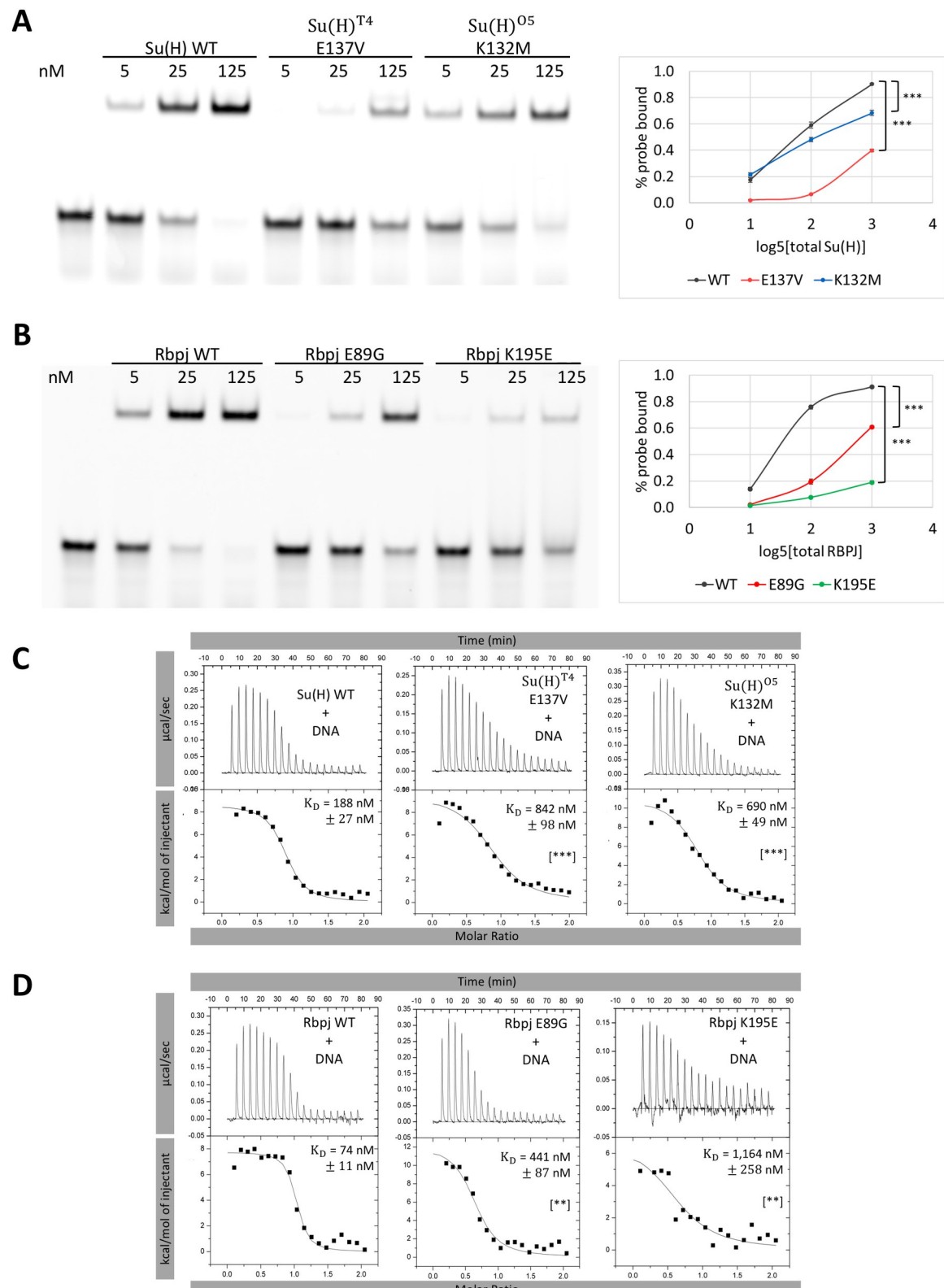

**Fig 2. Su(H) and RBPJ AOS variants decrease DNA binding. A-B.** Representative EMSAs with triplicate quantification shown on the right for purified (A) *Drosophila* Su(H) WT, E137V, and K132M proteins and (B) mouse Rbpj WT, E89G, and K195E proteins. P-values are reported for the 125 nM protein conditions and were determined by an ANOVA with Tukey Honest Significant Difference test ([***]

P < 0.001, [**] P < 0.01, [*] P < 0.05 and [NS] not significant). WT Su(H) vs. Su(H) K132M P = 2.14E-6. WT Su(H) vs. E137V P = 9.9E-9. WT RBPJ vs. E89G P = 2.27E-8. WT RBPJ vs. K195E P = 6.42E-14. **C-D.** Representative thermograms showing the raw heat signal and nonlinear least squares fit to the integrated data for (C) *Drosophila* Su(H) and (D) mouse Rbpj variants binding to a 20mer oligo duplex containing the same RBPJ/Su(H) binding site used in the above EMSAs. Each ITC experiment was performed at 10˚C with 20 injections. The mean dissociation constants ($K_D$)and standard deviations from triplicate experiments are reported along with the p-value determined from a two-tailed T-test ([***] P < 0.001, [**] P < 0.01, [*] P < 0.05 and [NS] not significant). WT Su(H) vs. Su(H) K132M P = 2.22E-4. WT Su(H) vs. E137V P = 8.22E-4. WT RBPJ vs. E89G P = 4.06E-3. WT RBPJ vs. K195E P = 3.97E-3.

from WT (**Fig 2C**, **S1 Table**). WT mouse Rbpj bound DNA with a 74 nM $K_d$, while the E89G and K169E variants had an ~6- and 16-fold reduction in binding, respectively (**Fig 2D**, **S1 Table**). These trends in reduced DNA binding are consistent with our EMSA data that similarly showed that the Su(H) E137V is more severe than K132M and Rbpj K169E is more adversely affected than E89G (**Fig 2A and 2B**). From these experiments, we can conclude that the effects of these variants on DNA binding are conserved between the fly and human proteins, suggesting that the known *Drosophila Su(H)* alleles can be used as a developmental model to study RBPJ variants that cause AOS in humans.

## The *Su(H)* AOS-like alleles exacerbate phenotypes associated with imbalanced Notch co-activator and Hairless co-repressor levels

Fortini and Artavanis-Tsakonas originally identified the $Su(H)^{T4}$ and $Su(H)^{O5}$ alleles in a genetic modifier screen of *Notch* pathway phenotypes [17]. In that study, they showed that flies heterozygous for these two *Su(H)* alleles altered Notch pathway phenotypes in sensitized backgrounds, and often in an opposing manner compared to flies that were heterozygous for *Su(H)* loss-of-function alleles. Hence, they classified these alleles as having "gain-of-function" activity. To better understand how each *Su(H)* allele impacts *Notch* and *Hairless* phenotypes in light of our new sequencing results, we quantitatively reanalyzed two well established *Notch* (*N*) sensitive phenotypes: wing nicking or "notches" that are found in *Notch* heterozygotes flies ($N^{+/-}$); and macrochaetae sensory bristles, which increase in number with lower *Notch* levels (i.e. $N^{+/-}$) but decrease in number with lower *Hairless* (*H*) co-repressor levels (i.e. $H^{+/-}$).

First, we directly compared the impact of having either a "gain-of-function" $Su(H)^{T4}$ or $Su(H)^{O5}$ allele or the "loss-of-function" $Su(H)^{IB115}$ allele encoding a non-functional protein due to a premature stop codon (K138-stop) [17,18]. Flies heterozygous for the $Su(H)^{IB115/+}$ null allele in an otherwise WT genetic background developed normal wings (no nicks observed, N = 174; **Fig 3A**). However, consistent with prior studies [17], we found that a small subset of flies heterozygous for either the $Su(H)^{T4}$ or $Su(H)^{O5}$ allele had a dominant wing nicking phenotype and quantitative analysis revealed that flies with the $Su(H)^{T4}$ allele had both a higher penetrance and increased wing nicking severity than the $Su(H)^{O5}$ allele (**Fig 3A and 3B**). Thus, consistent with our quantitative *in vitro* DNA binding assays (see **Fig 2**), we found that the $Su(H)^{T4}$ allele encoding the E137V variant has both a more dramatic impact on DNA binding and a stronger impact on wing nicking when compared to the $Su(H)^{O5}$ allele encoding the K132M variant.

To determine if the wing nicking phenotype in the more severe $Su(H)^{T4}$ allele could be suppressed by compensatory changes in the Notch pathway, we first used GFP-tagged BAC transgenes to add an extra copy of either WT *Su(H)* (*Su(H)-GFP*) or *Notch* (*Notch-GFP*) into flies heterozygous for the $Su(H)^{T4}$ allele. Importantly, increasing the gene dose of both *Su(H)* ($Su(H)^{T4}/+;Su(H)-GFP$) and *Notch* ($Su(H)^{T4}/+;Notch-GFP$) suppressed the wing nicking phenotype (**Fig 3A**). Similarly, we found that removing a single copy of the *Hairless* gene that encodes a co-repressor that specifically antagonizes Notch pathway activation, also suppressed wing nicking in $Su(H)^{T4/+}$ flies ($Su(H)^{T4/+};H^{1/+}$) (**Fig 3A and 3C**). Thus, the dominant wing

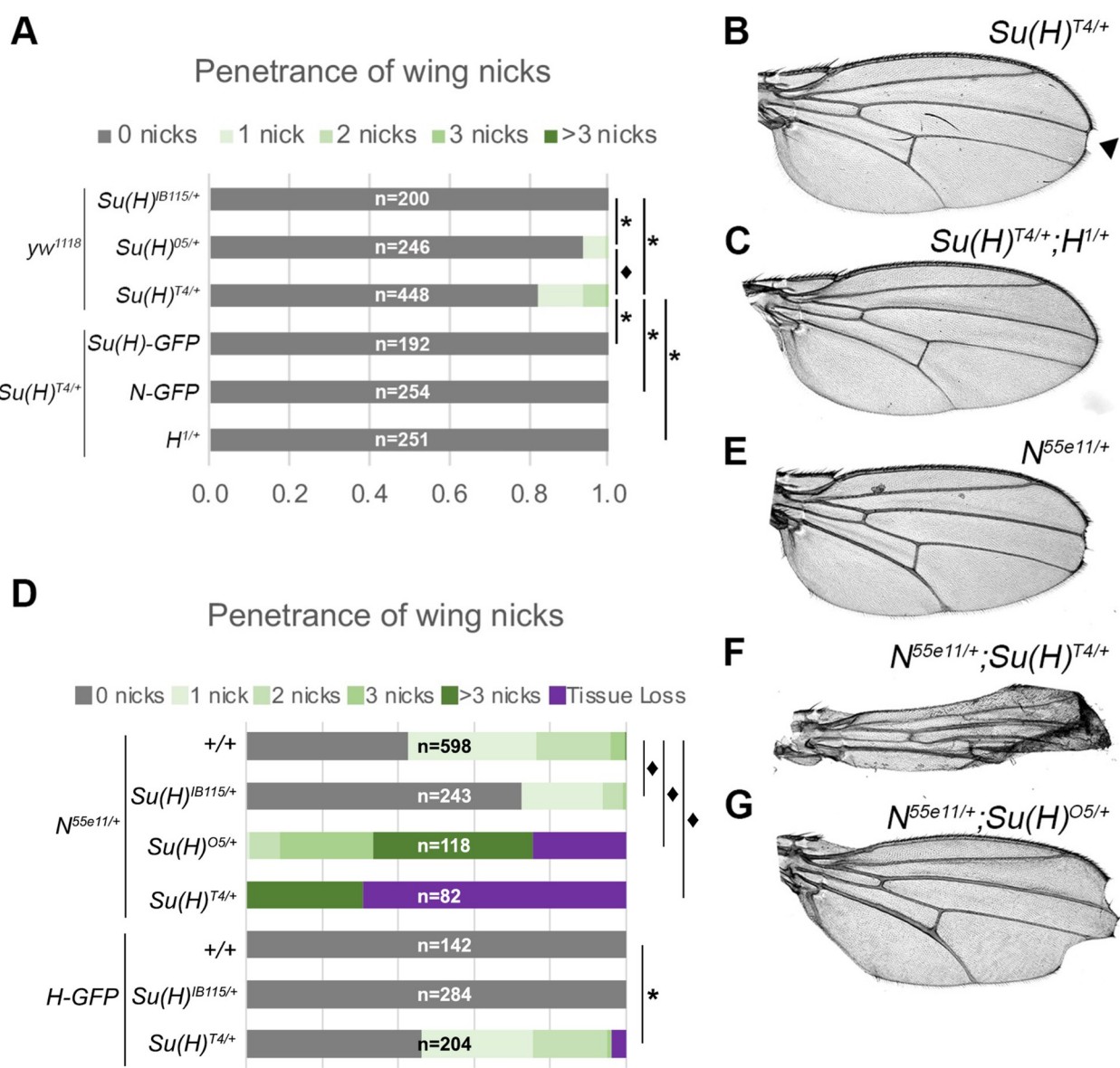

**Fig 3. AOS-like *Su(H)* alleles induce wing nicking phenotypes that can be enhanced or suppressed by genetic changes in the Notch signaling pathway.** **A.** Quantified wing nicking in the indicated genotypes with number of wings analyzed (n) noted on each bar graph. Flies used in the top three bars were generated by crossing the $yw^{1118}$ stock that is WT for *Su(H)* with flies carrying each of the indicated *Su(H)* alleles. Flies used in the bottom three bars were generated by crossing flies carrying a $Su(H)^{T4}$ allele with flies carrying either the *Su(H)-GFP* BAC, the *Notch-GFP* BAC, or the $H^1$ null allele, as indicated. Proportional odds model with Bonferroni adjustment tested for penetrance/severity differences between $Su(H)^{O5/+}$ and $Su(H)^{T4/+}$ flies with ♦ denoting p<0.01. Two-sided Fisher's exact test was used to assess all other genotypes with * denoting p<0.01. **B-C.** A wing from a $Su(H)^{T4/+}$ heterozygote in an otherwise WT background (B) or in combination with a $H^{1/+}$ heterozygous background ($Su(H)^{T4/+};H^{1/+}$, C). Note, the $Su(H)^{T4/+}$ wing had a nick (black arrowhead), whereas all the wings from $Su(H)^{T4/+};H^{1/+}$ flies did not have wing nicks. **D.** Quantified wing notching in the indicated genotypes with number of wings analyzed (n) noted on each bar graph. Flies used in the top four bars were generated by crossing $N^{55e11/+}$ female flies with either $yw^{1118}$ WT male flies or male flies carrying the indicated *Su(H)* alleles. Flies used in the bottom three bars were generated by crossing flies carrying the Hairless-GFP (H-GFP) BAC rescue with either $yw^{1118}$ WT flies or flies carrying the indicated *Su(H)* alleles. Two-sided Fisher's exact test was used to assess penetrance differences between flies with * denoting p<0.01. **E-G.** Wings from female flies containing a copy of the $N^{55e11}$ allele in either a WT (E), $Su(H)^{T4/+}$ (F) or $Su(H)^{O5/+}$ (G) background. Note, the severe nicking and morphological wing phenotypes observed in the compound heterozygotes.

phenotype caused by the $Su(H)^{T4}$ allele can be suppressed by either increasing the WT dose of $Su(H)$, increasing the gene dose of *Notch*, or decreasing the gene dose of the *H* co-repressor.

Next, we compared the ability of the $Su(H)^{T4}$, $Su(H)^{O5}$, and $Su(H)^{IB115/+}$ alleles to modify the wing nicking phenotype in *Notch* heterozygous animals that are known to have significant wing nicking in an animal with two WT $Su(H)$ alleles (**Fig 3D and 3E**). As expected, both the $Su(H)^{T4}$ and $Su(H)^{O5}$ alleles dramatically enhanced the wing notching phenotype in *Notch* heterozygous flies (i.e. $N^{55e11/+}$;$Su(H)^{T4/+}$ and $N^{55e11/+}$;$Su(H)^{05/+}$) with the $Su(H)^{T4}$ allele having a more dramatic impact on wing morphology (**Fig 3D, 3F and 3G**). In sharp contrast, the *Notch* haploinsufficiency wing phenotype was weakly, but significantly, suppressed in flies heterozygous for the $Su(H)^{IB115}$ null allele ($N^{55e11/+}$;$Su(H)^{IB115/+}$) (**Fig 3D**). Thus, while lowering the dose of $Su(H)$ using a null allele can partially alleviate the wing phenotype caused by too little *Notch*, the presence of the DNA binding compromised $Su(H)^{T4}$ and $Su(H)^{O5}$ alleles exacerbates this phenotype.

These data show that flies heterozygous for the $Su(H)^{T4}$ allele are susceptible to wing notching with a penetrance and severity that can be either enhanced in $N^{55e11}$ heterozygotes (**Fig 3D**) or suppressed by adding an extra *Notch* allele (*N-GFP* in **Fig 3A**). Similarly, we found that the ability of the $Su(H)^{T4}$ allele to induce wing nicking could also be suppressed by removing a *Hairless* allele (i.e. in $Su(H)^{T4/+}$;$H^{1/+}$ compound heterozygotes in **Fig 3A**). To further test how *Hairless* gene dose impacts the wing nicking phenotype induced by the $Su(H)^{T4}$ allele, we used a previously published *Hairless* BAC rescue transgene (H-GFP) [19] to add an extra *Hairless* allele in either wild type flies, $Su(H)^{T4/+}$ heterozygotes, or $Su(H)^{IB115/+}$ heterozygotes. Intriguingly, we found that flies with an extra *Hairless* allele developed normal wings in both a $Su(H)^{+/+}$ wild type background and in $Su(H)^{IB115/+}$ heterozygotes, whereas the extra *H-GFP* allele induced a dramatic increase in wing notching penetrance and severity in $Su(H)^{T4}$ heterozygotes (compare the $H$-$GFP$/$Su(H)^{T4/+}$ data in **Fig 3D** with the $Su(H)^{T4/+}$ data in **Fig 3A**). Taken together, these findings highlight how the gene dose of both *Notch* and *Hairless* can have a profound impact on the wing phenotypes generated by the $Su(H)^{T4}$ allele.

The wing nicking phenotype in *Notch* heterozygous flies is thought to be caused by decreased Notch signal strength in the Cut-positive wing margin cells during larval development [20–22]. To better understand the impact of the $Su(H)^{T4}$ and $Su(H)^{O5}$ alleles on Notch signal strength within these cells, we performed comparative expression analysis of the $E(spl)$ $m\alpha$-$GFP$ reporter gene in wing imaginal discs isolated from either WT, $Su(H)^{IB115}/+$, $Su(H)^{T4}/+$ or $Su(H)^{O5}/+$ larva. $E(spl)m\alpha$ is a known *Notch* target in the wing imaginal disc and is expressed in Cut+ wing margin cells as well as in additional cells that contribute to sensory organ development [23,24] (**Fig 4A**). To assess Notch signal strength in wing margin cells, we performed quantitative GFP expression analysis using confocal microscopy (see Methods for details) and used a Cut-antibody to specifically label wing margin cells. Since $E(spl)ma$-$GFP$ is activated very strongly in the sensory organ lineage that develops along the anterior wing margin by proneural transcription factors [25], we focused our analysis on the easily isolated and visualized posterior Cut+ margin cells (labeled posterior in **Fig 4A**, see diagram in **Fig 4A'** for location of wing margin cells and nearby sensory organs). Importantly, quantitative GFP analysis revealed that replacing a WT $Su(H)$ copy with the $Su(H)^{IB115}$ null allele did not significantly impact GFP levels, whereas both $Su(H)^{O5}$ and $Su(H)^{T4}$ heterozygotes had significantly lower GFP levels relative to WT and $Su(H)^{IB115}$ heterozygotes (**Fig 4B**). These data are consistent with the AOS-like $Su(H)^{T4}$ and $Su(H)^{O5}$ alleles, but not the $Su(H)^{IB115}$ null allele, being sufficient to lower Notch signal strength in wing margin cells, and thereby induce wing nicks.

Prior studies of flies heterozygous for either the $Su(H)^{T4}$ or $Su(H)^{O5}$ allele also found that both enhance the loss of macrochaetae sensory organs in flies heterozygous for the *Hairless* (*H*) co-repressor gene [17]. In contrast, and as its name implies, heterozygosity for a $Su(H)$

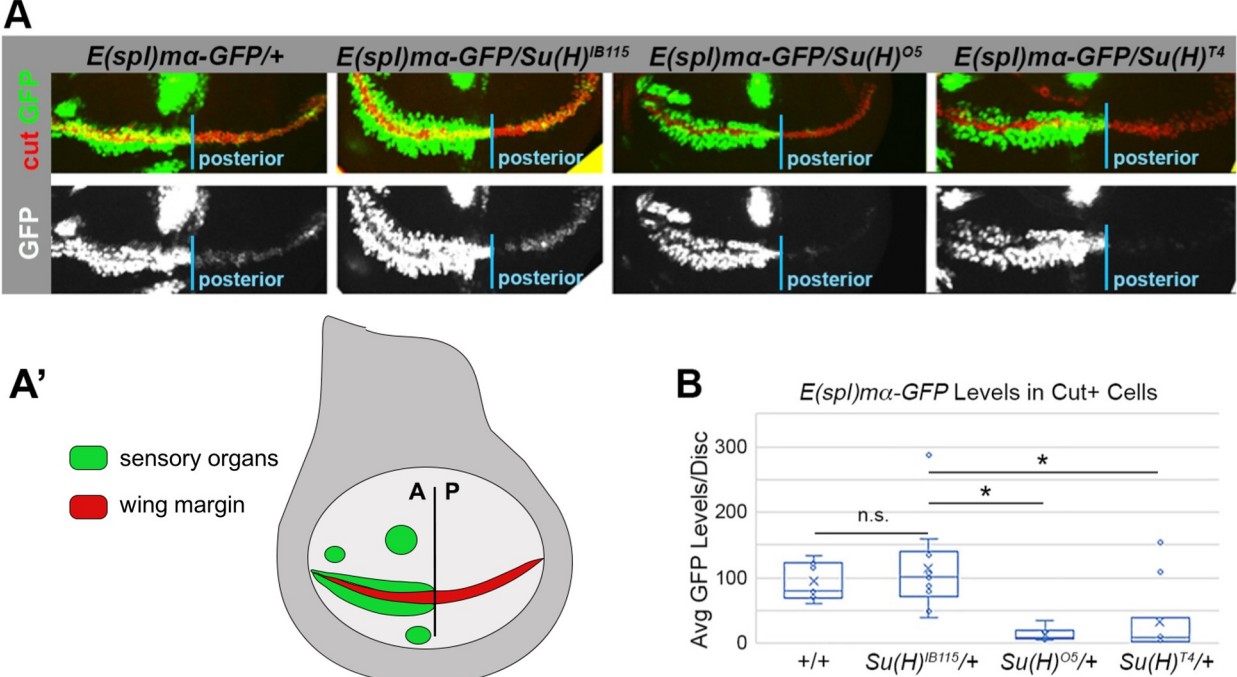

**Fig 4. Flies with an AOS-like *Su(H)* allele have dramatically lower expression of a *Notch* reporter gene in wing margin cells. A-A'**. Representative expression of the *E(spl)mα-GFP* reporter in larval wing discs from WT flies (far left) or from flies heterozygous for either *Su(H)^{IB115}*, *Su(H)^{O5}*, or *Su(H)^{T4}*, as indicated at top. Note, the wing discs were immunostained for Cut (red) to mark the wing margin cells and only the posterior margin cells were quantified in B. We focused on the posterior margin cells to avoid the strong GFP staining in the neighboring sensory organ cells because the *E(spl)mα-GFP* reporter is strongly activated by proneural transcription factors in a Notch-independent manner within the sensory organs [25]. Bottom panel shows GFP expression in grayscale. A'. Schematic of the larval wing disc with the wing margin cells (red) and the sensory organs (green) highlighted. Note, the absence of macrochaetae cells in the posterior compartment. **B**. Quantification of GFP levels from wing discs of the indicated genotypes. Each dot represents the average posterior wing margin cell pixel intensity as measured from an individual imaginal disc. * denotes significance (p<0.01) using a two-sided Student's t-test.

loss-of-function allele significantly suppresses the decrease in macrochaetae formation in *Hairless* heterozygous animals [26,27]. We similarly tested each of these *Su(H)* alleles in flies with either two WT alleles of *H* (i.e. a *yw^{1118}* stock) or in *H^{1/+}* heterozygotes. In the WT background, flies heterozygous for the *Su(H)^{IB115}* null allele invariably developed the expected 40 macrochaetae on the head and thorax of each adult fly (**Fig 5A**). Flies heterozygous for the *Su(H)^{O5}* and *Su(H)^{T4}* alleles also developed an average of 40 macrochaetae. However, unlike in *Su(H)^{IB115}* heterozygotes, we observed a small number of flies in both *Su(H)^{O5}* and *Su(H)^{T4}* heterozygotes that either had 1 or 2 missing macrochaetae or had 1 extra macrochaetae (**Fig 5A**). While the penetrance of these phenotypes is very low and the direction of change is variable (both increases and decreases in macrochaetae are observed), these data suggest that the process of macrochaetae selection and/or development may be less robust in *Su(H)^{O5}* and *Su(H)^{T4}* heterozygotes than in *Su(H)^{IB115}* heterozygotes.

Next, we tested each allele in the *H^{1/+}* heterozygotes that lose an average of ~12 macrochaetae (**Fig 5B–5F**). As expected, replacing a WT *Su(H)* allele with the *Su(H)^{IB115}* allele significantly suppressed the loss of macrochaetae in compound heterozygous animals (*Su(H)^{IB115}/+; H^{1/+}*, **Fig 5B and 5D**) [26,27]. In contrast, both the *Su(H)^{O5}* and *Su(H)^{T4}* alleles enhanced the loss of macrochaetae in *H^{1/+}* compound heterozygotes, and the *Su(H)^{T4}* allele again had a stronger impact than the *Su(H)^{O5}* allele (**Fig 5B, 5E and 5F**). Next, we analyzed the impact of each of these *Su(H)* alleles on macrochaetae phenotypes in flies sensitized to produce too

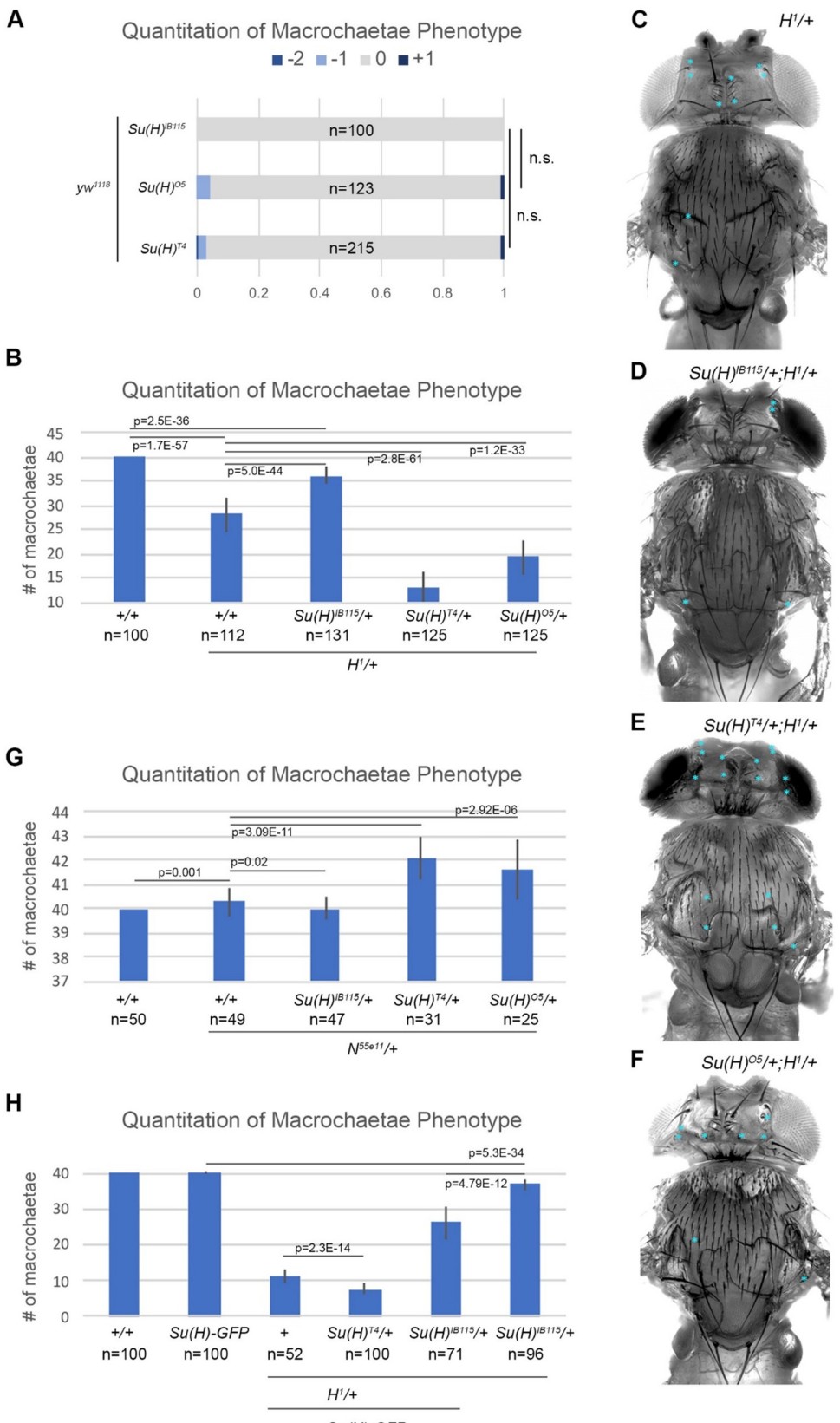

**Fig 5. AOS-like *Su(H)* alleles can enhance both macrochaetae loss or gain depending upon *Notch* and *Hairless* gene dose. A**. Quantified macrochaetae numbers in flies carrying either a single copy of *Su(H)*$^{IB115}$, *Su(H)*$^{O5}$, or *Su(H)*$^{T4}$, as indicated. Number of animals analyzed (n) noted on each bar graph. Note, flies heterozygous for the *Su(H)*$^{IB115}$ allele produce the expected invariable 40 macrochaetae on the head/thorax and show no variability. In contrast, a small percentage of both the *Su(H)*$^{O5}$ and *Su(H)*$^{T4}$ heterozygous flies either lost or gained a small number of macrochaetae. Significance was calculated using a two-sided Fisher's exact test. **B**. Quantified macrochaetae numbers in flies of the following genotypes: WT (+/+); *Hairless* heterozygotes (*H*$^{1/+}$); and *Su(H)/H*$^1$ compound heterozygotes with either the *Su(H)*$^{IB115}$, *Su(H)*$^{O5}$, or *Su(H)*$^{T4}$ alleles as indicated. Number of animals analyzed (n) noted below each genotype. Note, flies heterozygous for the *Su(H)*$^{IB115}$ allele suppress the *H*$^1$ loss of macrochaetae, whereas both *Su(H)*$^{O5}$ and *Su(H)*$^{T4}$ compound heterozygous flies enhance macrochaetae loss in a *Hairless* heterozygous background. Significance was calculated using a two-sided Student's t-test. **C-F**. Image of a *Drosophila* notum and head showing the loss of macrochaetae (marked by blue asterisks) in flies with the following genotypes: *H*$^{1/+}$ (C), *Su(H)*$^{IB115/+}$;*H*$^{1/+}$ (D), *Su(H)*$^{T4/+}$;*H*$^{1/+}$ (E); or *Su(H)*$^{O5/+}$;*H*$^{1/+}$ (F). **G**. Quantified macrochaetae numbers in flies of the following genotypes: WT (+/+); *Notch* heterozygotes (*N*$^{55e11/+}$); and *N*$^{55e11}$/*Su(H)* compound heterozygotes with either the *Su(H)*$^{IB115}$, *Su(H)*$^{O5}$, or *Su(H)*$^{T4}$ alleles as indicated. Number of animals analyzed (n) noted below each genotype. Note, flies heterozygous for the *Su(H)*$^{IB115}$ allele suppressed the *N*$^{55e11}$ gain of macrochaetae, whereas both the *Su(H)*$^{O5}$ and *Su(H)*$^{T4}$ heterozygous flies enhanced macrochaetae gain in a *Notch* heterozygous background. Significance was calculated using a two-sided Student's t-test. **H**. Quantified macrochaetae numbers in WT flies (+/+), flies carrying an extra copy of *Su(H)* (*Su(H)-GFP*), flies with an extra copy of *Su(H)* and heterozygous for *Hairless* (*H*$^1$/*Su(H)-GFP*), flies with *Su(H)*$^{T4}$/+;*H*$^1$/*Su(H)-GFP*, flies with *Su(H)*$^{IB115}$/+;*H*$^1$/*Su(H)-GFP*, and flies with *Su(H)*$^{IB115}$/+;*H*$^{1/+}$ as indicated. Number of animals analyzed (n) noted below each genotype. Significance was calculated using a two-sided Student's t-test.

many sensory bristles. *Notch* heterozygous flies produce a weakly penetrant, but significant increase in macrochaetae formation in animals with two WT alleles of *Su(H)* (**Fig 5G**). Intriguingly, we found that flies heterozygous for either the *Su(H)*$^{T4}$ or the *Su(H)*$^{O5}$ allele further enhanced the number of extra macrochaetae in *N*$^{55e11/+}$ flies (**Fig 5G**). In contrast, flies heterozygous for the *Su(H)*$^{IB115}$ null allele had the opposite effect and resulted in a small but significant suppression in the increase in macrochaetae in *N*$^{55e11/+}$ flies.

Taken together, these data support two ideas: First, even within the same tissue, the presence of either a *Su(H)*$^{T4}$ or *Su(H)*$^{O5}$ allele can significantly increase or decrease the number of sensory organs formed, but primarily only in sensitized genetic backgrounds that alter the amount of the Notch pathway co-activator (i.e. *Notch*) or co-repressor (i.e. *Hairless*) proteins. Second and in sharp contrast to the *Su(H)*$^{T4}$ or *Su(H)*$^{O5}$ alleles, we found that flies heterozygous for a *Su(H)* null allele (i.e. *Su(H)*$^{IB115/+}$) partially suppressed each of these phenotypes in the same genetic backgrounds. These data suggest that simply lowering the amount of Su(H) transcription factor can suppress imbalances in Notch co-activator to Hairless co-repressor levels, whereas having a DNA binding compromised Su(H) molecule enhances these imbalances to make phenotypes worse. To further investigate how *Su(H)* gene dose impacts sensory organ formation, we generated a series of flies containing an additional allele of WT *Su(H)* (i.e. *Su(H)-GFP* BAC). In an otherwise WT background, flies containing a third copy of *Su(H)* were phenotypically normal and did not change the number of macrochaetae (**Fig 5H**). However, in *Hairless* heterozygotes, adding an extra copy of *Su(H)* (i.e. *Su(H)*$^{+/+}$;*H*$^1$/*Su(H)-GFP*) enhanced the loss of macrochaetae (compare *H*$^{1/+}$ data in **Fig 5B** with the *H*$^1$/*Su(H)-GFP* data in **Fig 5H**), consistent with further increasing the levels of Su(H) transcription factor causing even stronger phenotypes. Next, we replaced one of the WT *Su(H)* alleles with the *Su(H)*$^{T4}$ allele (i.e. *Su(H)*$^{T4/+}$;*H*$^1$/*Su(H)-GFP*) and found that these flies had an even larger loss in macrochaetae (**Fig 5H**). In contrast, replacing one of the WT *Su(H)* alleles with the *Su(H)* null allele (i.e. *Su(H)*$^{IB115/+}$;*H*$^1$/*Su(H)-GFP*) resulted in flies with a similar number of macrochaetae as *H*$^{1/+}$ heterozygotes (**Fig 5G and 5H**). Altogether, these data suggest that the amount of Su(H) transcription factor is critical for proper Notch signaling and if one of these alleles encodes a Su(H) transcription factor with compromised DNA binding activity the phenotypes are dramatically enhanced, especially in flies with imbalanced levels of *Notch* and *Hairless*.

## Su(H) and RBPJ AOS variants bind normally to the NICD co-activator and the Hairless/SHARP co-repressor proteins

Integrating the DNA binding and genetic data suggests the following model: the $Su(H)^{T4}$ and $Su(H)^{O5}$ alleles produce proteins that sequester the NICD co-activator and Hairless co-repressor off DNA. Hence, flies heterozygous for either a $Su(H)^{T4}$ or $Su(H)^{O5}$ allele can exacerbate both wing notching and the gain in macrochaetae when Notch is limiting ($N^{55e11/+}$), and the loss in macrochaetae when Hairless is limiting ($H^{1/+}$). To further test this idea, we performed ITC assays to directly measure Su(H) and Rbpj binding to both co-activators and co-repressors. We hypothesized that the DNA binding mutations in Su(H)/RBPJ would not impact cofactor binding because these sites are distant from the interfaces of Su(H)/RBPJ involved in co-activator and co-repressor binding (**Fig 6A**). To test the Su(H) variants' ability to bind co-activators, we performed ITC with a construct that corresponds to the RAM domain of the *Drosophila* Notch receptor, which binds the β-trefoil domain (BTD) of Su(H) with high affinity [28] (**Fig 6A**). WT Su(H) bound to RAM with 187 nM affinity, whereas we observed no statistically significant differences in RAM binding to Su(H) E137V [*T4*] or K132M [*O5*] (**Fig 6B**, **S1 Table**). To test the Su(H) variants' ability to bind co-repressors, we performed ITC with Hairless, residues 232–358, which correspond to the region of Hairless that binds the C-terminal domain (CTD) of Su(H) with high affinity [29, 30] (**Fig 6A**). WT Su(H) bound Hairless with a 3 nM $K_d$ and we observed no significant differences in Hairless binding to Su(H) E137V or K132M (**Fig 6C**, **S1 Table**).

To test if mouse Rbpj proteins with the AOS variants showed similar trends with respect to co-activator/repressor binding, we performed ITC with the mouse RAM domain from Notch1 and the mouse co-repressor SHARP [31–33]. WT Rbpj and the E89G and K195E variants similarly bound RAM with an ~20 nM $K_d$ (**Fig 6D**, **S1 Table**). For Rbpj-SHARP complexes, WT and E89G bound the mouse SHARP protein with ~5 nM affinity, whereas K169E bound with a 21 nM $K_d$; however, this difference was not found to be statistically significant (T-test p value = 0.079). To further confirm that co-activator/repressor binding was unaffected for the mouse Rpbj protein containing analogous AOS variants, we performed coimmunoprecipitation experiments in HEK293 cells transfected with epitope-tagged versions (either GFP- or FLAG-tagged as indicated) of each Rbpj variant in the presence of either NICD1, NICD1 + MAML1, or SHARP (**Fig 6F–6H**). NICD1 coimmunoprecipitated with each mouse Rbpj AOS variants to a similar extent as WT Rbpj (**Fig 6F**) and MAML1 coimmunoprecipitated with NICD1 and Rbpj to similar levels with WT and variant Rbpj constructs (**Fig 6G**). Both mouse Rbpj AOS variants also coimmunoprecipitated equally well with SHARP compared to WT Rbpj (**Fig 6H**). Similarly, using a well-established mammalian two-hybrid assay that is independent of DNA binding by Rbpj, the variants E89G and K195E interact with the corepressor SHARP equivalently to WT Rbpj (**S3 Fig**). Taken together, these data support the hypothesis that the Su(H) variants and mouse Rbpj AOS variants specifically disrupt DNA binding without affecting co-activator/repressor binding, corroborating a model of cofactor sequestration.

## RBPJ AOS variants alter Notch-mediated transcription in mammalian cells via a sequestration mechanism

To confirm that the mouse Rbpj AOS variants do not impact protein subcellular localization, we transfected HeLa cells with GFP-tagged Rbpj (WT and variants) and monitored protein distribution using fluorescence microscopy. Like WT Rbpj, both AOS variants were primarily localized in the nucleus (**Fig 7A**). Additionally, we treated Rbpj transfected cells with cycloheximide and monitored protein half-life. The degradation rates of both Rbpj AOS variants were

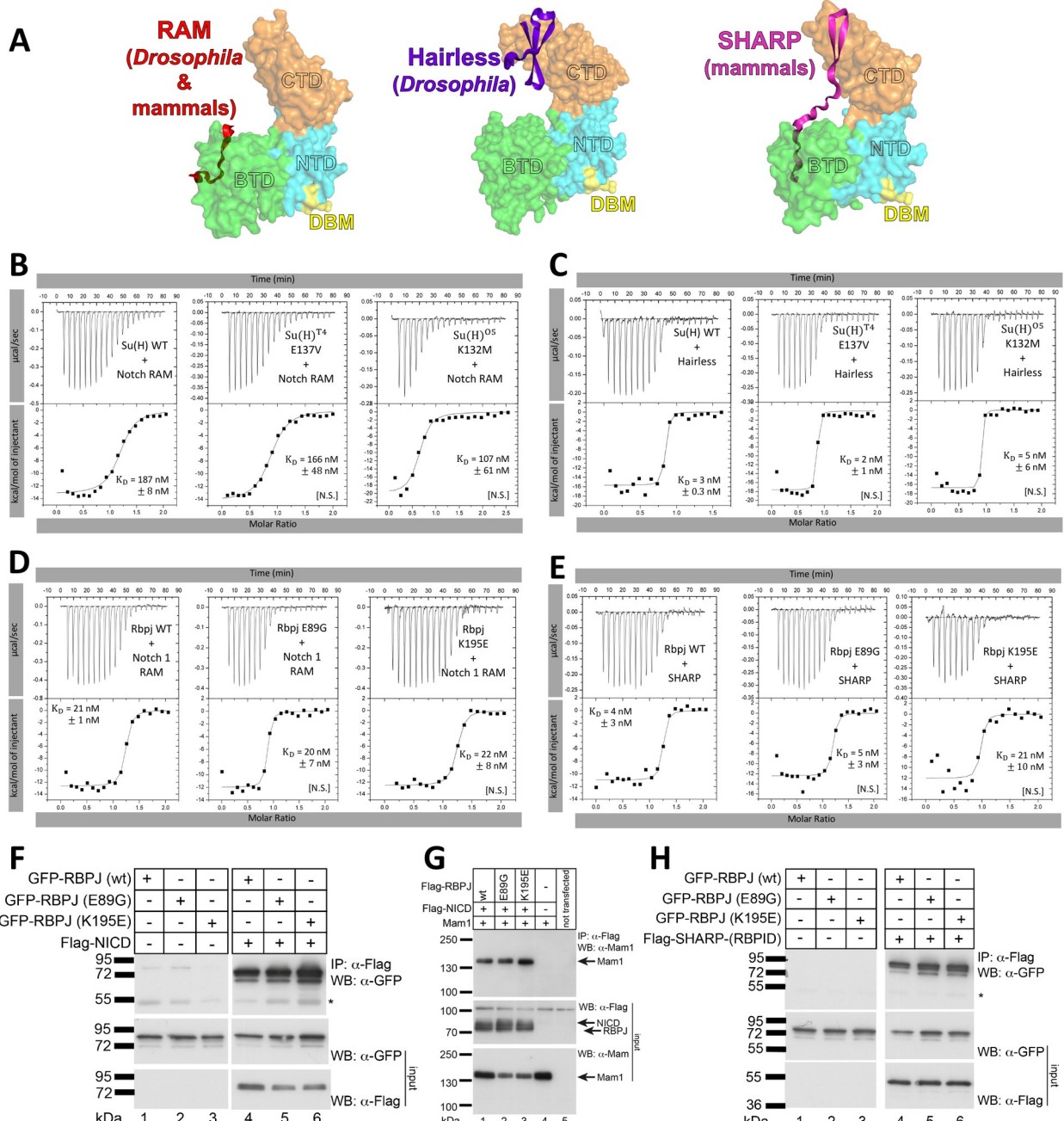

**Fig 6. The Su(H)/RBPJ variants bind the co-activator and co-repressor proteins as well as WT Su(H)/RBPJ. A.** Structures of the RAM domain (left, PDBID: 3V79) [51], Hairless (middle, PDBID: 5E24) [29], and SHARP (right, PDBID: 6DKS) [31] bound to Su(H)/RBPJ, with DNA removed from the model for simplicity. Su(H)/RBPJ is represented as a surface colored by domains as in Fig 1C, with the Su(H) and RBPJ DNA binding variants of the NTD colored yellow and labeled DBM (DNA Binding Mutation). The cofactors are represented as solid cartoons with RAM in red, Hairless in purple, and SHARP in pink. **B-E.** Representative thermograms showing the raw heat signal and nonlinear least squares fit to the integrated data from ITC experiments with (B) *Drosophila* Notch RAM in the syringe and Su(H) in the cell, (C) *Drosophila* Hairless in the syringe and Su(H) in the cell, (D) mouse Notch 1 RAM in the syringe and mouse Rbpj in the cell, and (E) mouse SHARP in the syringe and mouse Rbpj in the cell. Each ITC experiment was performed at 25°C with 20 injections. The average dissociation constants ($K_D$) and standard deviations from triplicate experiments are reported along with the p-value determined from a two-tailed T-test ([***] P < 0.001, [**] P < 0.01, [*] P < 0.05 and [NS] not significant). For the RAM binding experiments: WT Su(H) vs. Su(H) K132M P = 0.142. WT Su(H) vs. E137V P = 0.583. WT RBPJ vs. E89G P = 0.860. WT RBPJ vs. K195E P = 0.789. For the Hairless/SHARP binding experiments: WT Su(H) vs. Su(H) K132M P = 0.605. WT Su(H) vs. E137V P = 0.185. WT RBPJ vs. E89G P = 0.614. WT RBPJ vs. K195E P = 0.079. **F.** NICD binding of WT and variant mouse Rbpj proteins from cells. HEK293 cells were transfected with the plasmids GFP-Rbpj WT, GFP-Rbpj E89G, or GFP-Rbpj K195E in the absence or presence of Flag-NICD. Immunoprecipitation was performed with anti-Flag antibody agarose beads and detected by Western blotting using an anti-GFP antibody.

Expression of GFP-Rbpj (middle blot) was detected using an anti-GFP antibody. Expression of the Flag-NICD protein (bottom blot) was detected using an anti-Flag antibody. The asterisk in the upper blot marks the heavy chain of the antibody used for immunoprecipitation. **G.** Mouse Rbpj WT and Rbpj-AOS variants similarly form the NICD co-activator complex that includes Mam1. HEK293 cells were transfected with the plasmids expressing Maml1 in the absence or presence of Flag-Rbpj WT, Flag-Rbpj E89G, or Flag-Rbpj K195E and Flag-NICD. Co-immunoprecipitation was performed with the anti-Flag antibody agarose beads and detected by Western blotting using an anti-Maml1 antibody. Expression of Flag-Rbpj proteins (middle blot) and Flag-NICD was detected using an anti-Flag antibody. Expression of the Maml1 protein (bottom blot) was detected using the anti-Maml1 antibody. **H.** CoIP of mouse SHARP with WT and AOS variants of Rbpj. HEK293 cells were transfected with the plasmids GFP-Rbpj WT, GFP-Rbpj E89G, or GFP-Rbpj K195E in the absence or presence of Flag-SHARP-(RBPID, Rbpj-interaction domain). Immunoprecipitation was performed with the anti-Flag antibody agarose beads and detected by Western blotting using an anti-GFP antibody. Expression of GFP-Rbpj (middle blot) was detected using an anti-GFP antibody. Expression of the Flag-SHARP-(RBPID) protein (bottom blot) was detected using an anti-Flag antibody. The asterisk in the upper blot marks the heavy chain of the antibody used for immunoprecipitation.

similar to WT Rbpj (**S4 Fig**). We performed luciferase assays using a reporter containing 12 CSL binding sites in HeLa^RBPJ-KO cells, in which endogenous RBPJ had been deleted using CRISPR-Cas9 [34], to test whether the AOS variants could bind and activate the reporter in the presence and absence of NICD1 (**Fig 7B and 7C**). First, we expressed mouse Rbpj-VP16 fusion proteins, WT or AOS variants, to determine whether the AOS variants could bind and activate the reporter in a NICD1 independent manner. Both K195E and E89G Rbpj-VP16 fusions showed significant reduction in reporter signal compared to WT Rbpj-VP16 (**Fig 7B**), consistent with the above DNA binding experiments that show each variant protein has decreased DNA binding activity (**Fig 2**). To investigate the effect of these AOS variants on Notch-mediated transcription, we performed luciferase assays by co-transfecting with mouse NICD1 and Rbpj. In this case, we observed robust signaling with WT Rbpj and NICD1, whereas Rbpj K195E and E89G activated the reporter to a much lesser extent than WT. Moreover, it should be noted that the K195E variant showed a stronger negative impact on transcriptional responses when either fused to VP16 or co-transfected with NICD1 than the E89G variant, consistent with the K195E variant have a more dramatic impact on DNA binding.

Next, we assessed the ability of each Rbpj variant to repress target genes using a mature T (MT) cell line in which Notch signaling is in the "off" state. In this previously established system, CRISPR/Cas9-mediated deletion of Rbpj leads to upregulation of the Notch target genes *Hey1 and Hes1*, due to de-repression, but not the *Tbp* control, and this phenotype can be efficiently rescued by reintroducing WT mouse Rbpj expression via lentiviral transduction [31]. We generated WT Rbpj, E89G, and K195E MT cell lines and performed qRT-PCR to quantify each protein's ability to repress transcription at the *Hey1* and *Hes1* Notch target sites (**Fig 7D**). As expected, expression of WT Rbpj rescued repression *of Hey1* and *Hes1*, whereas the Rbpj K195E protein that more strongly affects DNA binding (**Fig 2**), failed to repress transcription at both *Hey1* and *Hes1* sites. However, Rbpj E89G, which has a lesser impact on DNA binding than K195E (**Fig 2**), restores repression at Hey1 and Hes1 similar to WT RBPJ. Taken together, these results suggest that the severity of the mutation on DNA binding directly corelates to the level of both transcriptional activation and repression.

Lastly, to test the sequestration mechanism in mammalian cells, we once again used our luciferase assays in HeLa^RBPJ-KO cells (**Fig 7E**). Because transfection of increasing amounts of WT Rbpj results in reduced reporter activity (**Figs 7E and S5B**), which has been reported previously [35,36] and is likely due to titration of NICD, we designed three additional Rbpj mutants: (1) Rbpj^KRS (K195E/R218H/S221D), which combines the variant K195E with two other previously described Rbpj mutants that affect DNA binding [37,38] and is completely devoid of any DNA binding (**S5 Fig**); (2) Rbpj^NBM (NICD Binding Mutant, F261A/V263A/R422E/E425R/E398R), which is unable to interact with NICD1 but retains its ability to bind DNA (**Figs 7E & S5**); and (3) Rbpj^NBM/KRS, which is neither able to bind DNA nor NICD1. It should also be mentioned that Rbpj^NBM has been tested in *Drosophila*, using the fly ortholog

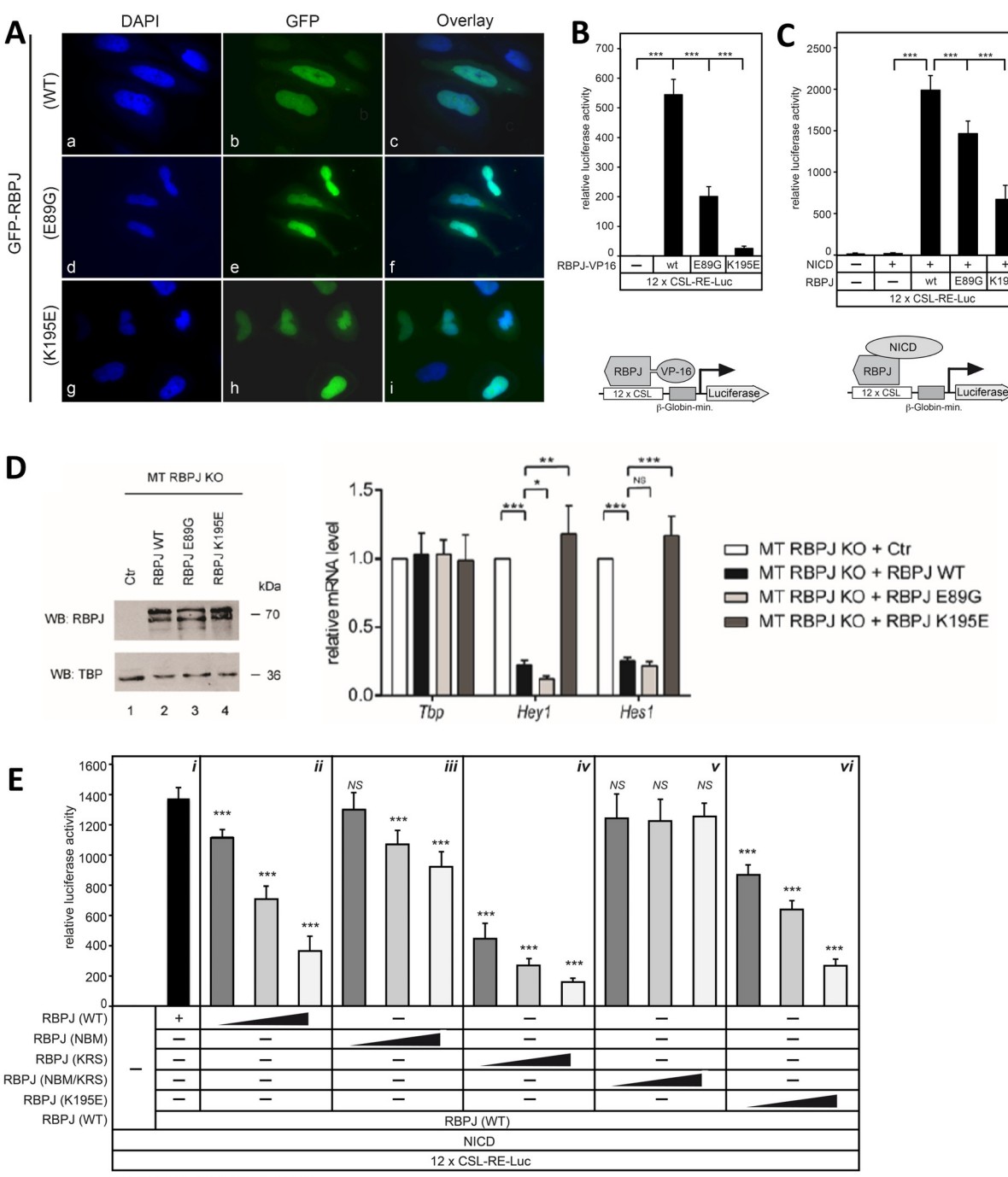

**Fig 7. The RBPJ AOS variants show dysregulated Notch target gene expression when compared to WT RBPJ. A.** Subcellular localization of Rbpj proteins after expression in HeLa cells, illustrating predominantly nuclear localization of both WT and variant Rbpj proteins. HeLa cells were transiently transfected with 0.3 μg of the respective GFP-Rbpj plasmids: WT, E89G, and K195E. After 24 hours, cells were fixed and stained with DAPI and imaged under a fluorescent microscope using a 63x objective. **B.** Luciferase reporter assay using an Rbpj-VP16 fusion to test for DNA binding capacities (schematic lower panel). Both Rbpj K195E-VP16 and E89G-VP16 show reduced reporter signal compared to WT, consistent with the observation that both have impaired DNA binding. HeLa^RBPJ-KO cells were transfected with the Notch/RBPJ dependent reporter 12xCSL-RE-Luc (250 ng) without or with the indicated Rbpj-VP16 constructs (10 ng). Luciferase activity was measured 24 hours after transfection. Bars represent mean values from six independent experiments; error bars indicate standard deviation, *** p< 0.001. **C.** Transactivation capacities of RBPJ WT and AOS variants together with NICD. HeLa^RBPJ-KO cells were transfected with the Notch/ RBPJ dependent reporter 12xCSL-RE-Luc (250 ng) without or with the indicated Rbpj constructs (50 ng) and the NICD expression plasmid (10 ng) (schematic lower panel). In RBPJ depleted HeLa cells, NICD is not able to transactivate the reporter. Together with NICD, the AOS

Rbpj variants show decreased transactivation capacities compared to Rbpj WT. Luciferase activity was measured 24 hours after transfection. Bars represent mean values from six independent experiments; error bars indicate standard deviation, *** p< 0.001. **D.** Rbpj-WT but not the AOS variant K195E are able to rescue transcription repression of Notch target genes. Left: Western blot against Rbpj (WT and variants) in reconstitution experiments using CRISPR/Cas9-mediated depletion of Rbpj in mature T (MT) cells. Rbpj protein expression levels were analyzed in Rbpj-depleted cells (control, line 1), Rbpj WT or Rbpj E89G and Rbpj K195E mutants, (lanes 2,3 and 4 respectively) and TBP was used as a loading control. Right: qRT-PCR: Expression of Rbpj WT and Rbpj E89G, but not Rbpj K195E rescue the repression of *Hey1* and *Hes1* Notch target genes. Total RNA was extracted from Rbpj depleted mature T cells reconstituted with either empty vector (control), Rbpj WT and Rbpj E89G or Rbpj K195E mutants. Data shown were normalized to the housekeeping gene *Bact* and represent the mean ± SD of three independent experiments ([***] P < 0.001, [**] P < 0.01, [*] P < 0.05 and [NS] not significant, unpaired Student's t-test). **E.** Luciferase reporter assays demonstrating that the cofactor sequestration mechanism occurs in mammalian cells. HeLa$^{RBPJ-KO}$ cells were transfected with the Notch/RBPJ dependent reporter 12xCSL-RE-Luc (250 ng), WT Rbpj (50 ng), and NICD (10 ng) as shown in panel *i*. Cells were additionally transfected with increasing concentrations (50, 150 & 250 ng) of either WT, NBM (F261A/V263A/R422E/E425R/E398R), KRS (K195E/R218H/S221D), NBM/KRS, or K195E Rbpj constructs in panels *ii-iv*. Rbpj NBM is unable to bind NICD and activate the reporter, and Rbpj KRS is completely ablated for DNA binding (S5 Fig). Increasing concentrations of Rbpj KRS strongly inhibit reporter activity due to a complete lack of DNA binding, but still able to bind cofactors (panel *iv*); however, increasing concentrations of Rbpj KRS/NBM have no effect on reporter activity (panel *v*) due to the inability of this construct to interact with NICD and titrate it off the reporter. (n ≥ 4, [***] P < 0.0001, [NS] not significant, unpaired Student's t-test).

Su(H), and in these *in vivo* assays was also shown to be completely lacking in NICD binding and Notch-mediated transcriptional activation [19]. For these experiments, a constant amount of WT Rbpj (50 ng) and NICD1 (10 ng) was co-transfected into cells to activate the reporter (**Fig 7E, panel *i***), and increasing amounts (50, 150, 250 ng) of either WT Rbpj (**Fig 7E, panel *ii***), Rbpj$^{NBM}$ (**Fig 7E, panel *iii***), Rbpj$^{KRS}$ (**Fig 7E, panel *iv***), Rbpj$^{NBM/KRS}$ (**Fig 7E, panel *v***), or the AOS variant K195E (**Fig 7E, panel *vi***) were additionally transfected into the cells. These experiments demonstrate the following: (1) Increasing cellular concentrations of the Rbpj DNA binding mutants KRS and K195E both significantly reduce reporter activity more than WT Rbpj, albeit the reduction by the AOS variant K195E is modest compared to the KRS mutant; (2) Increasing cellular concentrations of Rbpj$^{NBM}$, which cannot bind NICD1 but can still bind DNA, has only a very minor negative effect on reporter activity, likely by competing with WT Rbpj for DNA binding sites on the reporter; and (3) Increasing cellular concentrations of Rbpj$^{NBM/KRS}$, which cannot interact with NICD1 or DNA, has no effect on reporter activity. Taken together, these data provide proof-of-principle that Rbpj variants that are defective in DNA binding reduce Notch signaling via a cofactor sequestration mechanism.

## Discussion

Human genetic studies previously revealed that Adams Oliver Syndrome (AOS) is associated with autosomal dominant alleles in the *NOTCH1*, *DLL4*, and *RBPJ* genes [39]. Sequence analysis of these Notch pathway variants has led to the prediction that each variant is likely to compromise Notch signaling in specific tissues, and thereby cause a developmental syndrome that affects multiple organ systems. In knockout mouse studies, homozygous null *Notch1*, *Dll4*, or *Rbpj* mice are all embryonic lethal [40–42] and only *Dll4* heterozygous mice display a severe phenotype [12]. Here, we took advantage of the unexpected finding that a previously described dominant *Su(H)* allele in *Drosophila* [17] contains a missense variant in an analogous residue found in a family with AOS [14]. By combining quantitative genetic studies with quantitative DNA binding, protein-protein interaction, and transcriptional reporter assays, our findings provide evidence that Su(H)/Rbpj variants that compromise DNA binding can result in Notch target gene misregulation and phenotypes due to the sequestration of either the Notch signal or the antagonistic co-repressor proteins. Moreover, our findings reveal how increasing or decreasing the genetic dose of *Su(H)* can either suppress or exacerbate Notch pathway phenotypes depending on genetic background. Taken together with prior studies using these and similar *Su(H)* alleles as well as the human genetic data, our findings have several important

implications for better understanding the potential molecular defects underlying both AOS phenotypes and Notch pathway dysregulation.

First, our studies in *Drosophila* highlight how *Su(H)* allelic variants that specifically encode proteins with altered DNA binding, but not co-activator/co-repressor binding, behave differently than null *Su(H)* alleles that fail to generate any functional protein. In particular, we found that flies heterozygous for a *Su(H)* allele encoding a DNA compromised protein strongly enhances *Notch* and *Hairless* phenotypes, whereas simply lowering the genetic dose of *Su(H)* using a protein null allele partially suppresses these phenotypes. These findings support the idea that the Su(H)^T4 (E137V) and Su(H)^O5 (K132M) variants act as dominant negative proteins due to sequestering cofactors, especially in sensitized backgrounds where either Notch or Hairless are limiting. It should be noted that a similar mechanism was proposed and demonstrated by over-expressing a DNA binding deficient Rbpj (R218H) molecule in mammalian cell culture [43], which is also consistent with our cellular reporter assays using the Rbpj DNA binding mutants KRS and K195E (**Fig 7E**). This mechanism also likely explains the previously observed dominant gain-of-function activity of other described *Su(H)* alleles, such as *Su(H)^eBC11* (Y134N) [17,44] and *Su(H)^S5* (R266H) [27,38]. While the impact of the Su(H)^eBC11 Y134N variant on DNA and cofactor binding has yet to be experimentally tested, its location within the DNA binding domain and the reported dominant genetic behavior of the *Su(H)^eBC11* allele support the idea this variant is also likely to selectively compromise DNA binding. Moreover, the Su(H)^S5 R266H variant has been previously proposed to have dominant negative activity based on prior studies showing that it qualitatively disrupts DNA binding and was found to still interact with the Hairless protein in a yeast-two hybrid assay [45]. Our quantitative molecular and genetic studies further support these ideas and also help to explain how homozygous mutant clones of the *Su(H)^T4* allele cause a Notch loss-of-function phenotype [16]; namely, the loss of DNA binding by the Su(H)^T4 variant will result in the failure to robustly bind key target genes downstream of Notch activation. Thus, the *Su(H)^T4* allele should be viewed as both a dominant negative allele that can interfere with wild type Su(H) function in heterozygous cells by binding limiting co-factors, and a hypomorphic loss-of-function allele due to decreased DNA binding and the subsequent failure to activate key Notch target genes in homozygous mutant cells.

Interestingly, since the discovery of the AOS-associated E63G and K169E human RBPJ variants that were originally identified in 2012 [14], additional RBPJ AOS variants have been identified, including R65G, F66V, and S332R [39]. Importantly, none of the AOS-associated RBPJ alleles are predicted to be a protein null allele and given the role of R65 and F66 in DNA binding, as shown in previous Rbpj structures [33], and their close proximity to E63 in the NTD of RBPJ, these missense variants will also likely reduce DNA binding, while having little to no effect on cofactor binding. S332, which is currently classified as a variant of unknown significance [46], does not directly contact DNA, but rather is located in a long β-strand that connects the BTD and CTD of RBPJ. However, it is possible that the S332R AOS variant will incur structural changes that indirectly decrease the overall binding affinity for DNA, resulting in a similar disease mechanism. Nonetheless, we have established a rigorous platform to test these and future AOS variants *in vitro* and *in vivo*.

Intriguingly, an analogous cellular mechanism that compromises Su(H) DNA binding has previously been identified in *Drosophila* [37,47]. In this case, phosphorylation of Su(H) residue S269, which interacts with DNA, blocks DNA binding, and plays a role in fly hematopoiesis. Moreover, it is tempting to speculate that the severity of AOS disease is potentially linked to the impact the variant has on DNA binding. Certainly, our genetic studies in *Drosophila* support this premise, i.e. the *Su(H)^T4* allele encoding the E137V variant had a stronger impact on DNA binding in our *in vitro* assays than the *Su(H)^O5* allele encoding the K132M variant (**Fig**

**2**), and the $Su(H)^{T4}$ allele consistently led to stronger mutant phenotypes *in vivo* (**Figs 3–5**). Perhaps some of the patient variability can be explained by the severity of the RBPJ variant; however, there are likely other genetic and environmental modifiers that also contribute to an individual's disease presentation, obscuring the relative effect a variant has on DNA binding.

Second, our studies, as well as the original publication that identified the $Su(H)^{T4}$ and $Su(H)^{O5}$ alleles [17], revealed strong genetic interactions with not only *Notch*, but also the antagonistic *Hairless* co-repressor gene. These findings raise the possibility that human AOS-associated RBPJ alleles with compromised DNA binding may cause developmental defects due to the sequestration of mammalian co-repressor proteins and not simply due to sequestration of the Notch signal. However, it should be noted that repression mechanisms are not as well conserved across species, as there is no direct Hairless ortholog in mammals and mammalian RBPJ can bind multiple different co-repressors [10]. Moreover, we found that only the stronger AOS variant Rbpj K195E was defective in the MT cell repression assay (**Fig 7D**), whereas Rbpj E89G, which has a milder defect in DNA binding, functioned similarly to WT in this repression assay. Additionally, no known variants in Notch pathway co-repressor genes have been associated with AOS, whereas numerous *NOTCH1* and *DLL4* variants have been associated with this developmental syndrome [39], and several of the autosomal dominant *NOTCH1* and *DLL4* AOS alleles encode nonsense variants that are unlikely to generate a protein with dominant-negative effects. Thus, the clinical similarities between AOS patients with variants in NOTCH1, DLL4, and RBPJ suggest that the majority of developmental defects associated with RBPJ variants that encode proteins with compromised DNA binding are likely due to decreased NOTCH1 signal strength and not due to an impact on the antagonistic co-repressors. Nevertheless, since the genetic etiology of human patients with AOS is far from complete, future studies may want to include sequence analysis for variants in co-repressor genes known to interact with RBPJ.

Third, our *Drosophila* genetic data generated by systematically testing the impact of both increasing and decreasing *Notch* and *Hairless* gene dose in the presence of different *Su(H)* alleles highlight how genetic background can dramatically sensitize tissue-specific phenotypes. For example, flies heterozygous for the $Su(H)^{T4}$ allele only have a weakly penetrant wing notching phenotype in a genetic background with two wild type alleles of *Notch* and *Hairless*. However, simultaneously lowering the *Notch* gene dose or increasing the *Hairless* gene dose dramatically enhanced the wing phenotype, whereas increasing *Notch* gene dose or decreasing *Hairless* gene dose suppressed the wing phenotype. Moreover, we similarly found that while the $Su(H)^{T4}$ allele on its own did not dramatically impact macrochaetae development, simultaneously changing the relative dose of *Notch* and *Hairless* could significantly exacerbate macrochaetae phenotypes in compound heterozygotes with the $Su(H)^{T4}$ allele. Collectively, these data suggest that genetic modifiers in the presence of the dominant-negative AOS-like *Su(H)* allele can have tissue-specific impacts based on whether they either strengthen or weaken the Notch co-activator complex versus Hairless co-repressor complex. Lastly, we non-intuitively found that simply changing the gene dose of the *Su(H)* transcription factor can modify (i.e. partially suppress) each of these Notch- and Hairless-dependent phenotypes. Currently, we do not have a good molecular explanation for how decreasing the expression levels of the common transcription factor could suppress both Notch- and Hairless-dependent phenotypes, and future experiments are needed to define the complex relationships that exist between *Notch*, *Hairless*, and *Su(H)* gene dose in each specific tissue.

Fourth, our studies and proposed sequestration disease mechanism raise an interesting unanswered question—since RBPJ is the sole downstream nuclear effector of Notch signaling, why do AOS patients harboring RBPJ variants have distinct and nonoverlapping phenotypes with Alagille syndrome patients that have variants in NOTCH2 or JAG1 [15]? Current data

suggests that the NOTCH signal (i.e. the intracellular domains) generated by both NOTCH1 and NOTCH2 interact with RBPJ with similar affinities [48,49], suggesting that the DNA binding compromised RBPJ proteins should similarly sequester the signals generated by either NOTCH1 or NOTCH2. Hence, it is unclear why the AOS-associated RBPJ alleles that cause similar phenotypes as the AOS-associated *NOTCH1* variants do not also cause Alagille-like phenotypes that have been associated with *NOTCH2* variants that cause decreased NOTCH2 activity. While answering this question is beyond the scope of this study, developing vertebrate models of AOS will certainly help address this question and provide further insights into the molecular mechanism of AOS, Alagille, and Notch signaling in general.

## Methods

### Bacterial expression constructs

Su(H) amino acid residues 98–523 corresponding to the conserved and structurally ordered core domain were cloned into the pGEX-6P1 bacterial expression vector [28]. Based on previous studies [30], additional mutations R155T and N218G, which are surface exposed and distal from sites of DNA and co-regulator binding, were introduced to improve upon protein purification, yield, and stability. These mutations were previously shown to have identical binding to Notch, Hairless, and DNA as WT Su(H) [30]. Quick-change site directed mutagenesis was used to introduce the mutations K132M and E137V into the pGEX-6P1-Su(H) construct. For the mammalian ortholog, mouse Rbpj amino acids 53–474, corresponding to the structural core, were cloned into pGEX-6P1 and the E89G and K195E variants were introduced as mentioned above [33].

### Mammalian expression constructs

The expression plasmids pcDNA3-Flag-hsNotch-1-IC (Flag-NICD), pcDNA3-Flag-Rbpj (WT), pcDNA3-Rbpj-VP16(WT), pcDNA-3.1-Flag1-hsSHARP (2770–3127), pFA-CMV-MINT and the luciferase reporter construct pGa981/6 (12 x CSL-RE-LUC) were previously described [31]. The expression plasmids for AOS specific mouse Rbpj variants, pcDNA3-Flag-Rbpj(E89G), pcDNA3-Flag-Rbpj(K195E), pcDNA3-Rbpj-VP16(E89G) and pcDNA3-Rbpj-VP16(K195E) were made by site directed mutagenesis. For the Rbpj specific mutants KRS (K195E / R218H / S221D), NBM (F261A / V263A / R422E / E425R / E398R) and KRS/NBM the cDNAs were commercially synthesized (BioCat, Heidelberg) as SacII / XhoI fragments and inserted into the corresponding sites of pcDNA3-Flag-Rbpj(WT) and pcDNA3-Rbpj-VP16 (WT). All constructs were validated by sequencing.

### Bacterial recombinant protein expression and purification

As previously described [28,33], competent BL21 Tuner cells transformed with pGEX-6P1-Su (H) or pGEX-6P1-Rbpj were grown at 37˚C in LB + ampicillin to an OD of 1.5 followed by overnight IPTG induction at 20˚C. Cell pellets were resuspended in lysis buffer, sonicated, and centrifuged at 15,000 g. 60% w/v ammonium sulfate was added to the supernatant to precipitate the protein and then centrifuged at 15,000 g. The protein pellet was resuspended in PBS + 0.1% triton and incubated overnight with glutathione affinity resin. GST-Su(H) or -RBPJ tagged protein was eluted from the column and then cut with PreScission Protease. An SP ion exchange column was used to separate the cut GST from the Su(H) or RBPJ protein, which was then concentrated, flash frozen in liquid nitrogen and stored at -80˚C. Drosophila Hairless residues 232–358 and human SHARP residues 2776–2833, corresponding to the regions that

bind Su(H) and Rbpj, respectively, were cloned into pSMT3 to produce recombinant protein with an N-terminal SMT3 and His tag and purified as previously described [29–31].

## Peptide synthesis

*Drosophila* Notch residues 1763–1790 (VLSTQRKRAHGVTWFPEGFRAPAAVMSR) and human Notch1 residues 1754–1781 (VLLSRKRRRQHGQLWFPEGFKVSEASKK) corresponding to the fly and mouse RAM domains were chemically synthesized and purified to 95% purity by Peptide2.0.

## Electrophoretic Mobility Shift Assays (EMSAs) using purified proteins

EMSAs using bacterially purified proteins were performed essentially as previously described [52–54]. In brief, a fluorescent labeled probe containing a high affinity CSL binding site (shown in bold) was generated by annealing an unlabeled 5'–CGAACGAGGCAAACC-TAGGCTAGAGGCAC**CGTGGGGAA**ACTAGTGCGGGCGTGGCT– 3' oligonucleotide with a 5'IRDye-700 labeled 5'–AGCCACGCCCGCACT– 3' oligonucleotide that is complementary to the underlined region of the longer oligonucleotide. The Klenow enzyme was used to make the double-stranded probe and the indicated amounts of each Su(H) and Rbpj proteins were incubated with 3.4 nM of the labeled probe for 10 minutes at room temperature prior to being loaded onto native acrylamide gel electrophoresis. Acrylamide gels were run at 150V for 2 hours and imaged and quantified using the LICOR Odyssey CLx scanner.

## In vitro protein translation

The in vitro protein translations were performed using the TNT-assay (L4610) from Promega according to manufacturer's instructions. Prior to EMSAs the in vitro translations of Rbpj (WT) and mutant proteins were monitored by western blotting using an anti-Flag antibody (M5, Merck).

## EMSA from in vitro protein translation system

Reticulocyte lysates from in vitro translations were used for electromobility shift assays (EMSAs) in a binding buffer consisting of 10 mM Tris-HCl (pH 7.5), 100 mM NaCl, 0.1 mM EDTA, 0.5 mM DTT, and 4% glycerol. For the binding reaction, 10 ng (0.02 U) poly(dI-dC) (GE healthcare) and approximately 0.5 ng of $^{32}$P-labeled oligonucleotides were added. The sequence of the double-stranded oligonucleotide with 2 RBPJ binding sites (underlined): 5′-CCT GGA ACT ATT TTC CCA CGG TGC CCT TCC GCC CAT TTT CCC ACG AGT CG-3 and reverse strand: 5′-CTC GCG ACT CGT GGG AAA ATG GGC GGA AGG GCA CCG TGG GAA AAT AGT TC-3′. Super shifting of complexes was achieved by adding 1 μg of anti-Flag (M5, Merck) antibody. The reaction products were separated using 5% polyacrylamide gels with 1x Tris-glycine-EDTA at room temperature. Gels were dried and exposed to X-ray films (GE Healthcare).

## Oligonucleotide Preparation for ITC

The following 20-mer oligonucleotide sequences were ordered from Eurofins: 5'-GGCAC**CGTGGGA**AACTAGTG-3' and the reverse compliment 5'-CACTAGTT**TCC-CACG**GTGCC-3' showing the Su(H)/RBPJ binding site in bold and underlined. Single stranded oligos were further purified on a Resource Q ion exchange column and then buffer exchanged into an oligo annealing buffer containing 10 mM Tris 8.0, 500 mM NaCl, and 1

mM MgCl$_2$. Single stranded oligos were combined in equal molar amounts, boiled for 10 minutes, and then slowly cooled to room temperature to allow for proper annealing.

## Isothermal titration calorimetry

All proteins, peptides, and oligos were dialyzed overnight in a buffer containing 50 mM sodium phosphate and 150 mM sodium chloride. Experiments were conducted using the VP-ITC MicroCalorimeter manufactured by MicroCal. All protein/DNA binding experiments were performed at 10˚C, while protein/peptide experiments were performed at 20˚C. Each experiment was performed in triplicate with 20 injections, 14 μL per injection. Heat of dilution experiments were performed by injecting syringe samples into a cell containing only buffer, and all analysis was performed with the heat of dilution subtracted before fitting. The raw data were analyzed using ORIGIN and fit to a one site binding model. A two sample T-test was used to compare WT proteins to each mutant, with a p-value $<0.05$ indicating a significant difference.

## Differential scanning fluorimetry

Rbpj/Su(H) samples were prepared in triplicate at 5 μM in buffer containing 50 mM sodium phosphate and 150 mM sodium chloride. DSF was performed by heating samples containing 10X SYPRO Orange in 1˚C increments from 25˚C to 100˚C using the StepOne real-time PCR machine and using the Protein Thermal Shift software (ThermoFisher) to fit the data.

## Genomic DNA isolation and sequencing

For the original determination of the mutations in the *Su(H)*$^{T4}$ and *Su(H)*$^{O5}$ alleles, chromosomal DNA was isolated by protease K digestion of fly lysates. Genomic DNA was subjected to PCR with primers specific to the Su(H) coding region. Due to the presence of a large intron in the coding region, the first exon was amplified and sequenced separately using two primers TTGCAGCCTTAAACAGAAGCCAGC (forward) and AGCCGGTATTAT-CAGGTGCTTGGT (reverse). The resulting PCR fragment was sequenced using the same primers as well as an internal primer, ACAAATGCAGATGTCCTTGCTGCC. Exons 2–4 were amplified using two primers, CAAAGCTGCATTGCTTGCGGTT (forward) and TCAATCTACAAACTAAGGTCTTCG (reverse). The resulting PCR fragment was sequenced using these primers as well as two additional primers (ACAGTCAAACTGGTGTGCTCCG-TAA and ATGTAGAAGGCGCATTTGTGCAGC). Since the original fly stocks were heterozygous for the *Su(H)*$^{T4}$ and *Su(H)*$^{O5}$ alleles, the mutations were identified by the presence of double peaks in the resulting chromatogram. To confirm specific coding variants in each allele prior to our genetic phenotyping assays, genomic DNA was prepared using Qiagen's DNeasy blood purification kit. 10 adult males were used per genotype. DNA was eluted in 100ul Buffer AE. We used two primers (5'-GTCCAGTCCGCAATGAAAAT-3' (forward) and 5'-TGCTGCAACATCTCCTCGTA-3' (reverse)) to cover approximately 300bp surrounding each point mutation. The same primers were used for DNA sequencing and ExPasy.org was used to translate nucleotide to amino acid sequence. Note, the sequencing of these *Su(H)* alleles were previously reported to Flybase via personal communication by Dr. Patrick Dolph [44].

## Fly husbandry

All flies were cultured on standard cornmeal food at 25˚C. The following alleles were obtained from the Bloomington Drosophila Stock Center: Su(H)$^{T4}$(#63234), Su(H)$^{O5}$(#63235), Su

(H)$^{IB115}$ (#26661), N$^{55e11}$(#28813), H$^1$(#515), E(spl)mα-Gal4, UAS-GFP (#26788), PBac[Su (H)-GFP,FLAG] (#81279) and PBac[N-GFP,FLAG] (#81271). The Hairless::GFP BAC flies were a kind gift from Dr. Sarah Bray [19].

## Reporter assay in wing disc

Homozygous E(spl)mα-Gal4, UAS-GFP male flies were mated to either WT, Su(H)$^{IB115}$/cyo, act-GFP, Su(H)$^{T4}$/cyo,act-GFP or Su(H)$^{O5}$/cyo,act-GFP virginal females. Imaginal wing discs from non-cyo-act-GFP wandering 3$^{rd}$ instar larvae were dissected in PBS and fixed in 4% formaldehyde for 15min. Discs were subsequently washed 4 times with PBX (0.3% TritonX-100 in PBS), and incubated with the primary antibody, anti-Cut (Mouse 1:50, DSHB). No antibody was used for GFP detection. A fluorescent secondary antibody (Goat anti-Mouse 555 Alexa Fluor, Molecular Probes) was used to detect the Cut-positive wing margin cells. For quantitative purposes, all imaginal discs were dissected, fixed, and imaged at the same time on a Nikon A1R inverted confocal microscope (40x objective) with identical exposure settings. Z-stack images were analyzed using Imaris software. A Student's t-test was used to determine significance.

## Genetic assays

We analyzed the wing nicking and macrochaetae phenotypes essentially as previously described [52]. In brief, flies of the appropriate genotypes were mated and transferred to fresh food every other day to avoid overcrowding. Offspring of the listed genotypes were selected and the number of nicks on each wing and/or the number of macrochaetae on the dorsal head and thorax was recorded. Proportional odds model with Bonferonni adjustment and two-sided Fisher's exact test were used to determine significance for wing nicking. Student's t-test and two-sided Fisher's exact test were used to determine significance for bristle number.

## Cell culture and preparation of cell extracts

Rbpj-depleted hybridoma mature T (MT) cell line was grown in Iscove's Modified Dulbecco Medium (IMDM, Gibco) supplemented with 2% FCS, 0.3 mg/l peptone, 5 mg/l insulin, nonessential aminoacids and penicillin/streptomycin. Cells were grown at 37˚C with 5% CO$_2$. Phoenix packaging cells (Orbigen, Inc. San Diego, CA, USA) were cultivated in Dulbecco's Modified Eagle Medium (DMEM, Gibco) supplemented with 10% fetal calf serum (FCS) and penicillin/streptomycin. CRISPR/Cas9-mediated *Rbpj* depleted hybridoma mature T cells were generated as previously described [31]. Rbpj-depleted mature T-cell lines stably expressing Rbpj$^{WT}$, Adams-Oliver mutant Rbpj$^{E89G}$ or Adams-Oliver mutant Rbpj$^{K195E}$ were generated as follows: 5 x 10$^6$ Phoenix cells were transfected with 20 μg of the retroviral plasmid DNA mixed with 860 μl of H$_2$O and 120 μl of 2x HBS buffer (50 mM HEPES pH 7.05, 10 mM KCl, 12 mM Glucose, 280 mM NaCl, 1.5 mM NaHPO$_4$) while vortexing and the solution was incubated 20 min at room temperature. In the meantime, 25 μM Chloroquine solution (Sigma-Aldrich) was added to the Phoenix cells (1 μl/ml) and the cells were incubated for 10 min. The DNA solution was added to the cells and 12 h later the medium was replaced. After 24 h of incubation, the medium containing the retroviral suspension was filtered and Poly-brene (Sigma-Aldrich) solution was added. Fresh medium was added to the Phoenix cells that were maintained in culture for further infections. The retroviral solution was used for spin infection of Rbpj-depleted MT cells by centrifuging 45 min at 1800 rpm at 37˚C. In total, four spin infections were performed over two days. Positively infected cells were selected with Blasticidin (Gibco).

## Preparation of protein extracts and western blotting from Rbpj-depleted mature T cells

The nuclear extract (NE) from MT cells overexpressing the Rbpj constructs or control cells containing empty vector was prepared as follows. Briefly, 10 x 10$^6$ cells were washed with PBS and resuspended in 200 μl of Buffer 1 (10 mM HEPES pH 7.9, 10 mM KCl, 0.1 mM EDTA, 0.1 mM EGTA, 1mM beta-mercaptoethanol, supplemented with PMSF). The cell suspension was incubated 10 min on ice, 5 μl of 10% NP-40 were added and mixed by vortexing. After 10 s of centrifugation at 13000 rpm at 4˚C, the nuclei pellet was washed twice in 500 μl of Buffer 1 and resuspended in 100 μl of Buffer 2 (20 mM HEPES pH 7.9, 400 mM NaCl, 1 mM EDTA, 1 mM EGTA, 1 mM beta-mercaptoethanol, supplemented with PMSF). After 20 min of incubation on ice, the nuclei suspension was centrifuged 10 min at 13000 rpm at 4˚C. and the supernatant was collected for further analysis. Protein concentration was measured by Bradford assay (Biorad) and samples were boiled after adding SDS-polacrylamide gel loading buffer. Samples were resolved by SDS-Page and analyzed by Western blotting using antibodies against RBPJ (Cosmo Bio Co. LTD) or TBP (Santa Cruz Biotechnology).

## Gene expression analysis as measured by qRT-PCR in hybridoma mature T-cells

Total RNA was purified using TRIzol reagent accordingly to manufacturer's instructions. 1 μg of RNA was reverse transcribed into cDNA using random hexamers and M-MuLV reverse transcriptase (New England Biolabs). Quantitative PCRs were assembled with Absolute QPCR ROX Mix (Thermo Scientific, AB-1139), gene-specific oligonucleotides and double-dye probes and analyzed using StepOne Plus Real Time PCR system (Applied Biosystem). Data were normalized to the housekeeping gene Beta actin (*Bact*) and represent the mean ± SD of three independent experiments (*** $p<0.001$; ** $p<0.01$, * $p<0.05$ and [ns], not significant, unpaired Student's t-test).

## Coimmunoprecipitation experiments

HEK293 and HeLa cells were transfected using the Profectin and Lipofectamine 2000 transfection reagent, respectively, according to the manufacturer's instructions. HEK293 cells were transfected with the indicated constructs for expression of untagged, GFP- and Flag-tagged WT and mutant proteins. 24 hours after transfection cells were lysed with 600 μl CHAPS lysis buffer [10 mM 3-[(3-Cholamidopropyl)dimethylammonio]-1-propanesulfonate hydrate (CHAPS, Roth), 50 mM Tris-HCl (pH 7.8), 150 mM NaCl, 5 mM NaF, 1 mM Dithiothreitol (DTT, Merck), 0.5 mM Phenylmethanesulfonyl fluoride (PMSF, Merck) and 40 μl/ml "Complete Mix" protease inhibitor cocktail (Roche)]. The extracts were incubated with 40 μl agarose-conjugated anti-Flag antibody (M2, Sigma) at 4˚C overnight. Precipitates were washed 6 to 8 times with CHAPS lysis buffer and finally resuspended in SDS-polyacrylamide gel loading buffer. For western blotting the proteins were resolved in SDS-polyacrylamide gels and transferred electrophoretically at room temperature to PVDF membranes (Merck) for 1 h at 50 mA using a Tris-glycine buffer system. The membranes were pre-blocked for 1 h in a solution of 3% milk powder in PBS-T (0.1% Tween 20 in PBS) before adding antibodies. The following antibodies were used: anti-GFP (7.1/13.1, mouse monoclonal IgG, secondary antibody peroxidase conjugated sheep anti-mouse IgG, NA931V, GE healthcare) or anti-Flag (M5, Sigma; secondary antibody, NA931V, GE healthcare). Anti-Mam1 (ab155786), rabbit polyclonal, Abcam, (1:1000), overnight, 4˚C, secondary: HRP-linked-donkey-anti-rabbit IgG (NA934V), GE-healthcare, (1:5000), 1h, rt.

## Fluorescence microscopy

HeLa cells were cultured on glass coverslips at a density of $10^5$ cells per cm$^2$. After 16 h cells were transfected with 300 ng of expression plasmids using the Lipofectamine 2000 transfection reagent. 24 h after transfection cells were rinsed with PBS, fixed with 4% paraformaldehyde (PFA, Merck) in PBS (pH = 7.5). Specimens were embedded in "ProLong Gold antifade" reagent (Thermofisher) supplemented with 2-(4-carbamimidoylphenyl)-1H-indol-6-carboximidamide (DAPI) and stored at 4˚C overnight. Pictures were taken using a fluorescence microscope (IX71, Olympus) equipped with a digital camera (C4742, Hamamatsu), and a 100-W mercury lamp (HBO 103W/2, Osram). The following filter sets were used: Green, (EGFP) ex: HQ470/40, em: HQ525/50, blue (DAPI) D360/50, em: D460/50.

## Luciferase assays

HeLa cells and HeLa$^{RBPJ-KO}$ cells were seeded in 48-well plates at a density of $2.5 \times 10^5$ cells. Transfection was performed with Lipofectamine 2000 reagent (ThermoFisher) using 0.5 μg of reporter plasmid alone or together with various amount of expression plasmids (given in the corresponding figure legends). After 24 h luciferase activity was determined from at least four independent experiments with 20 μl of cleared lysate in a Centro LB 960 Microplate luminometer (Berthold) by using the Dual-Glo Luciferase Assay System from Promega.

## Cyclohexamide assays

Mouse mK4 cell line was maintained in DMEM/F12 media supplemented with 10%FBS, penicillin, and streptomycin. Transfection of 6xMyc-tagged RBPJ constructs (pCS2-RBPJ-6xMyc, WT and AOS variants) was carried out with TransIT-293 transfection reagent (Mirus) following the manufacture's instruction. After 24 h of the transfection, the cells were treated with cycloheximide (500ng/ml) and harvested at 48, 72, and 96 h. Protein samples were subjected to SDS-PAGE and subsequently transferred to PVDF membranes. The following primary antibodies were used at 1:1000 dilution in PBST (0.05% Tween-20) and 5% non-fat dry milk and incubated overnight at 4˚C: Myc (Cell Signaling Technology #2276), RBPJ (Cell Signaling Technology #5313), and actin (Cell Signaling Technology #5125). Protein bands were visualized by SuperSignal West Pico PLUS enhanced chemiluminescent reagent (ThermoFisher) and detected by ChemiDoc MP system (BioRad).

## Supporting information

**S1 Table. Calorimetric Binding Data for Native and Variant RBPJ/Su(H) Proteins.**
(DOCX)

**S1 Fig. Differential scanning fluorimetry of purified WT and variant Su(H) (top) and RBPJ (bottom) proteins.** The mean melting temperature (Tm) and standard error were calculated from triplicate experiments.
(TIFF)

**S2 Fig. EMSA from quick coupled *in vitro* transcription/translation system (TNT).** Full length Flag-Rbpj WT (lane 1 and 2) binds to DNA in contrast to Flag-Rbpj K195E mutant (lane 5 and 6) and Flag-Rbpj E89G mutant (lane 3 and 4), which interact weakly with DNA. Protein-DNA interaction of Flag-Rbpj WT is shown by complex A (single occupancy, lane 1 and 2), complex A' (double occupancy, lane 1 and 2) and complex B and B' with the addition of anti-Flag showing supershifting (lane 2). Bottom panel: Western blot to show similar

expression levels of WT and variant Rbpj constructs.
(TIFF)

**S3 Fig. Mammalian two-hybrid assay to analyze WT and Rbpj AOS variants binding to SHARP in cells.** HeLa cells were cotransfected with the indicated Gal4-SHARP (300ng) and Rbpj-VP16 (50ng) constructs together with the pFR-Luc reporter (500ng), which contains five Gal4 DNA binding sites upstream of the luciferase gene. Relative luciferase activity was determined after cotransfection of the pFR-Luc reporter construct alone. WT, E89G, and K195E activate the reporter similarly, suggesting that the AOS variants bind SHARP similar to WT Rbpj, whereas the Rbpj double-mutant F261A/L388A, which is compromised for SHARP binding, does not activate the reporter.
(TIFF)

**S4 Fig. Cycloheximide chase analysis of Myc-tagged RBPJ and RBPJ variant proteins. MK4 cells were transfected with Myc-tagged RBPJ constructs (human wild-type RBPJ, E63G, and K169E) followed by cycloheximide treatment (0.5ug/ml) for 24, 48, and 72 hours.** Ectopic expression of Myc-tagged RBPJ constructs was detected by western blot analysis with anti-Myc (9B11) and anti-RBPJ antibodies. The signal intensities of corresponding protein bands were quantified by ImageJ software and expressed as ratios relative to those at day 1.
(TIFF)

**S5 Fig. Subcellular localization and luciferase reporter assays of AOS Rbpj variants. A.** Subcellular localization of Rbpj proteins after expression in HeLa cells, illustrating predominantly nuclear localization of both WT and variant Rbpj proteins. HeLa cells were transiently transfected with 0.3 μg of the respective GFP-Rbpj plasmids: WT, E89G, K195E, NBM (NICD binding mutant: F261A/V263A/R422E/E425R/E398R), KRS (DNA binding mutant: K195E/R218H/S221D) or NBM/KRS. After 24 hours, cells were fixed and stained with DAPI and imaged under a fluorescent microscope using a 63x objective. **B.** Luciferase reporter assays using different levels of WT Rbpj expression. HeLa$^{RBPJ-KO}$ cells were co-transfected with the Notch/RBPJ dependent reporter 12xCSL-RE-Luc (250 ng), without or with NICD (10 ng), and Rbpj (5, 10, 50, 100, 200, 300 ng). Luciferase activity was measured 24 hours after transfection. Maximal reporter activity was observed with 50 ng of transfected Rbpj DNA. Bars represent mean values from six independent experiments and error bars indicate standard deviation. ([**] $P < 0.001$, [***] $P < 0.0001$, unpaired Student's t-test). **C.** Luciferase reporter assays demonstrating that Rbpj mutants NBM and KRS are unable to bind NICD and DNA, respectively. HeLa$^{RBPJ-KO}$ cells were transfected with the Notch/RBPJ dependent reporter 12xCSL-RE-Luc (250 ng) without or with the indicated Rbpj or Rbpj-VP16 constructs (50 ng), and with or without NICD (10 ng). Luciferase activity was measured 24 hours after transfection. While Rbpj-VP16 NBM is able to activate the reporter, addition of NICD does not result in an addition increase in reporter activity due to Rbpj NBM unable to bind NICD. Rbpj-VP16 KRS and NBM/KRS are unable to activate the reporter with or without NICD. Bars represent mean values from six independent experiments and error bars indicate standard deviation. ([***] $P < 0.0001$, [NS] not significant, unpaired Student's t-test).
(TIFF)

## Acknowledgments

We thank the *Drosophila* stock centers and Dr. Sarah Bray for fly stocks and the Developmental Studies Hybridoma Bank (DSHB) for antibody reagents. We also thank Sabine Schirmer and Roswitha Rittelmann for excellent technical assistance.

## Author Contributions

**Conceptualization:** Brian Gebelein, Rhett A. Kovall.

**Data curation:** Ellen K. Gagliani, Lisa M. Gutzwiller, Rhett A. Kovall.

**Formal analysis:** Ellen K. Gagliani, Lisa M. Gutzwiller, Yoshinobu Odaka, Phillipp Hoffmeister, Stefanie Hauff, Aleksandra Turkiewicz, Emily Harding-Theobald, Patrick J. Dolph, Tilman Borggrefe, Franz Oswald, Brian Gebelein, Rhett A. Kovall.

**Funding acquisition:** Tilman Borggrefe, Franz Oswald, Brian Gebelein, Rhett A. Kovall.

**Investigation:** Ellen K. Gagliani, Lisa M. Gutzwiller, Yi Kuang, Yoshinobu Odaka, Phillipp Hoffmeister, Stefanie Hauff, Aleksandra Turkiewicz, Emily Harding-Theobald, Patrick J. Dolph, Tilman Borggrefe, Franz Oswald, Brian Gebelein, Rhett A. Kovall.

**Methodology:** Ellen K. Gagliani, Lisa M. Gutzwiller, Yi Kuang, Yoshinobu Odaka, Phillipp Hoffmeister, Stefanie Hauff, Aleksandra Turkiewicz, Emily Harding-Theobald, Patrick J. Dolph.

**Project administration:** Patrick J. Dolph, Tilman Borggrefe, Franz Oswald, Brian Gebelein, Rhett A. Kovall.

**Resources:** Patrick J. Dolph, Tilman Borggrefe, Franz Oswald, Brian Gebelein, Rhett A. Kovall.

**Supervision:** Patrick J. Dolph, Tilman Borggrefe, Franz Oswald, Brian Gebelein, Rhett A. Kovall.

**Writing – original draft:** Ellen K. Gagliani, Lisa M. Gutzwiller, Yoshinobu Odaka, Patrick J. Dolph, Tilman Borggrefe, Franz Oswald, Brian Gebelein, Rhett A. Kovall.

**Writing – review & editing:** Ellen K. Gagliani, Lisa M. Gutzwiller, Yi Kuang, Patrick J. Dolph, Tilman Borggrefe, Franz Oswald, Brian Gebelein, Rhett A. Kovall.

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
