## [Decision Letter · Decision Letter 0]

15 Nov 2021

Dear Dr Kovall,

Thank you very much for submitting your Research Article entitled 'A Drosophila Su(H) Model of Adams-Oliver Syndrome Reveals Notch Cofactor Titration as a Mechanism Underlying Developmental Defects' to PLOS Genetics.

The manuscript was fully evaluated at the editorial level and by three independent peer reviewers. The reviewers appreciated the attention to an important problem, but raised some substantial concerns about the current manuscript. Based on the reviews, we will not be able to accept this version of the manuscript, but we would be willing to review a much-revised version. We cannot, of course, promise publication at that time.

If you decide to revise the manuscript for further consideration at PLOS Genetics, please aim to resubmit within the next 60 days, unless it will take extra time to address the concerns of the reviewers, in which case we would appreciate an expected resubmission date by email to plosgenetics@plos.org.

[LINK]

We are sorry that we cannot be more positive about your manuscript at this stage. Please do not hesitate to contact us if you have any concerns or questions.

Yours sincerely,

Pablo Wappner

Associate Editor

PLOS Genetics

Gregory P. Copenhaver

Editor-in-Chief

PLOS Genetics

Reviewer #1: Two disease-causing dominant variants of human CSL, a key transcription factor transducing the Notch signal, were recently identified in AOS patients. These variants map to the DNA-binding domain of CSL and impair sequence-specific DNA-binding. This indicated that these AOS mutations cause a loss of Notch signaling activity. How CSL mutations resulting in a loss of Notch activity cause a dominant effect in patients is not clear.

Here, the T4 mutation of fly CSL (E137V) was shown to change the same conserved residue as the one mutated in one of the two AOS variants (E63G). Interestingly, this mutation was originally classified as a gain-of-function (gof) allele. Another previously identified gof allele of fly CSL (O5, K132M) was also mapped to its DNA-binding domain. These two mutations are shown here to impair the in vitro DNA-binding activity of fly CSL.

These observations led the authors to study how a loss of DNA-binding activity can result in a gain of activity. Detailed genetic interaction assays confirmed earlier observations showing that the T4 and O5 alleles dominantly enhance the phenotypes resulting from a partial loss of activity of either Notch or Hairless, a Notch antagonist. As nuclear Notch and Hairless interact directly with fly CSL, or Su(H), the authors interpret these counter-intuitive results to suggest that the DNA-binding defective CSL proteins act dominantly by titrating co-factors. In support of this, both T4 and O5 mutant proteins were shown to interact in vitro with NICD and Hairless.

The T4 and O5 alleles were also associated with a weak dominant wing margin phenotype and with reduced level of expression of a Notch reporter at the wing margin. Therefore, consistent with the Notch haplo-insufficient phenotype, nuclear Notch appears to be limiting during wing margin development.

The above interpretation for the T4 and O5 alleles was then extended to the AOS alleles of human CSL. Indeed, results from cells transfected with AOS mutant proteins fused to VP16 indicated that AOS mutant proteins have reduced DNA-binding activity. Moreover, the AOS mutants were shown to physically interact with both NICD/MAML and SHARP, a CSL co-repressors. Finally, transfection studies indicated that both trans-activation by Notch and basal repression are altered in AOS variants with impaired DNA-binding activity.

Together, these data strongly suggest that these fly and human mutant CSL factors exert their dominant effect by titrating both positively- and negatively-acting factors.

This is very nice paper. The data are clear and convincing. The conclusion is important as it provides a clear molecular interpretation for the dominant effect of the AOS mutations of human CSL. It also provides a rationale to further design tools to search for limiting partners of CSL in various cellular contexts.

Minor comments:

1. it would be nice to show that the T4 and O5 alleles indeed behave as loss-of-function alleles in the absence of wild-type Su(H) (see point 2 below).

2. two other gof alleles of Su(H) have been described. The eBC11 mutation (Y134N, see Flybase) maps very close to the O5 and T4 mutations studied here and might therefore affect DNA-binding. The S5 mutation (R266H, see: The Drosophila homolog of the immunoglobulin recombination signal-binding protein regulates peripheral nervous system development. Furukawa, T., Maruyama, S., Kawaichi, M., and Honjo, T. Cell 69, 1191-1197 [1992]) was previously shown to impair DNA-binding (see: Suppressor of hairless, the Drosophila homologue of RBP-Jk, transactivates the neurogenic gene E(spl)m8. Furukawa, T., Kobayakawa, Y., Tamura, K., Kimura, K., Kawaichi, M., Tanimura, T. and Honjo, T. Jpn. J. Genet. 70, 505-524 [1995]). Therefore, the main conclusion of the current study is likely to apply to these two alleles. This could be discussed.

In further support of this piece of work, the T4 and S5 alleles were previously suggested to behave as loss-of-function alleles when homozygous in mosaic genetics (Schweisguth, F., and Lecourtois, M. Genes Dev Evol 208, 19-27 [1998]). This observation led the authors to suggest that these mutant proteins may act by titrating Hairless. This could also be discussed.

3. Typo (p 11): (compare H1/+ data in Fig 5B with the H1/S(H)-GFP data in Fig 5H)

Reviewer #2: This is a well constructed study that serves to connect previous observations from flies and patients in a clear way. The data are all of high quality and show clearly that a specific set of mutations in the transcription factor Su(H)/RBPJ affect DNA binding, leading to a model that these mutated proteins can alter Notch activity by titrating either the active NICD co-factor or the corepressor. One aspect that could be examined more is the extent that corepressor titration is a driver for phenotypes in an otherwise normal background. To date most of the data suggest that the effects on NICD predominate (e.g. the wing nick in the flies and the interpretations from the patient data). Second, as the work builds on previous observations many of the results are not novel. For example, the fly data largely recapitulate the original findings of Fortini et al, albeit the analysis in the current manuscript has been performed in a much more thorough and well quantified way. In this framework it is hard to see the major step forward from the work, although it may be helpful, as stated above, for connecting the dots between the fly and the mammalian studies.

The starting point for the work is the fact that a class of mutations affecting the human RBPJ are potentially analogous to some previously identified Drosophila Su(H) alleles. Sequencing of patients with Adams-Oliver syndrome (AOS) had identified mutations in RBPJ that were proposed to be loss of function mutations, based on the likelihood that DNA binding was compromised. In contrast, the Drosophila alleles have some unusual characteristic, that differed from a deletion of the locus and in some assays behaved in the opposite manner, leading to a gain of function classification. In particular, the alleles enhanced the phenotype from a reduction in the co-repressor Hairless. The most parsimonious explanation is the one presented here, that the mutations interfered with DNA binding and the Su(H) protein formed would then sequester the NICD and co-repressors in a poorly functioning complex. In agreement the mutated residue in one of the first allele (Su(H)[S5]) to be sequenced affects the DNA binding domain. However, this model was not formally presented before and in this manuscript the authors give a clear framework and provide data that support the model with a set of very thorough and careful biochemical and genetic analyses.

The data here confirm the alleles produce defects in DNA binding, using EMSA and ITC, and retain binding to their cofactors: both the fly Su(H) mutants and the mutated human RBPJ retain the ability to bind NICD and the co-repressors (Hairless or SHARP) based on IPs and ITC studies. These data nicely illustrate the similarities between the fly and human variants. The genetic data in flies recapitulate previous findings in a very clear way, to support the titration model. However, the human cell data illustrate the loss of function effects (reduced target gene expression) without demonstrating that it is due to the mechanism inferred. Given the historical background that has already established much of the framework for this study, a further step is needed to close the loop by showing the titration mechanism is causal in human cells.

Major points

1. The authors have performed classic genetic interactions with Notch alleles and Hairless alleles that in essence recapitulate the findings from previously (e.g. Fortini et al, Ashburner) The data are much better explained and properly quantified here, but the findings are not novel, except for those showing the effects of increased Notch ( Notch-GFP BAC) that agree with the titration model. To make a step forward they could test other predictions from the model to make a more complete study. e.g. what do they predict from an extra copy of Hairless? for phenotypes from NICD expression? Or from combinations with an allele of Su(H) that has reduced Hairless binding?

2. Mammalian experiments do not really test the model, they validate the loss of DNA binding. Extending the mammalian cell experiments to show evidence in support of the titration model is needed.

3. The wing nick phenotypes from [T4] and [O5] resemble Notch loss of function rather than Hairless, why do Notch loss of function effects predominate over those from titrating Hairless? Is this of relevance for the mammalian condition, can they demonstrate that there is titration of SHARP in mammalian cells?

4. Why does Su(H)[lof]/+ suppress the Notch “nick” (loss-of-function) phenotype (Fig 3D)? In most previous studies, reducing the Su(H) dose has not modified the phenotype. Doesn't simple logic predict that, if anything, the Notch loss of function phenotype would be enhanced?

5. Why is there no effect on Cut, a target of Notch that is needed for wing margin? And also no affect an anterior E(spl)malpha? The story would be strengthened by extending the expression analysis to other genetic combinations, where the model predicts stronger defects.

Minor points:

6. The genetics in the background papers are opaque but the authors need to be cautious in their explanation about the Su(H) “gain-of-function” alleles. The basis on which these alleles are gain of function is the enhancement of the phenotype from reduced Hairless. It’s important to note that the alleles are lethal over Su(H) deletions or null alleles and so might be considered loss of function at the same time. In this respect the observations are not 100% at odds with the mammalian data, although they do hint at a more complex explanation.

7. The sequence of the T4 and O5 mutants has been published in Flybase since 2010. The work here confirms the results. The fact that this is a confirmatory result and the data have been in the public domain for 11 years should be made clear.

8. There are other Su(H) alleles with similar properties, including Su(H)[S5], sequenced by Furukawa et al (1992). The R266H mutation has subsequently been shown to affect DNA binding. This fits with the current study, but they should acknowledge that at least one member of this class of Su(H) alleles was already known to affect DNA binding.

9. The images in Figure 4 are not very clear. At minimum a diagram with the positions of the macrochaetae would be helpful.

Reviewer #3: Gagliani et al study mutant versions of Drosophila Su(H) and mouse Rbpj modeled after point mutations in human RBPJ implicated in the Adams-Oliver syndrome. Unlike other AOS mutations in NOTCH1 and DLL4, which seem to be nulls, RBPJ AOS mutants are all missense mutations. The authors use purified proteins and biophysical methods to show that AOS-modelled Su(H)/Rbpj mutants have reduced binding to DNA, but retain normal affinity for interaction with the Notch icd coactivator and the H and SHARP corepressors. They proceed to use in vivo assays to gauge the functionality of these variants in flies and mammalian cultured cells. The combined in vitro and in vivo data suggest that these mutants act as dominant negatives both in target activation (because they sequester Notch icd off DNA) and in target repression (because they sequester H or SHARP off DNA). The experiments are well performed and described and convincingly support the model proposed.

Besides some small improvements (itemized below), I would like to see a more thorough discussion of the fly and mouse Su(H)/Rbpj genetics from the existing literature (they already have a good coverage of the AOS human genetics). In other words, since the authors make the point that these mutants behave differently than the null alleles (like Su(H)IB115), they should discuss what is known about the null alleles in these two model organisms. Any dominant (null/+) or recessive (homozygous null) phenotypes should be listed for comparison with the new data presented here. I suppose that, unlike T4 in Drosophila, AOS-like point mutants do not exist for mice, but the knockout has surely been studied extensively. Indeed, why do the herein reported dominant negative alleles behave oppositely (and not simply more severely) than null alleles? The null Su(H) allele (IB115) dominantly suppresses both N/+ wing notching and H/+ bristle loss (both of which are enhanced by the T4 and O5 alleles). If wing integrity relies on the amount of Su(H)/Nicd complex and bristle specification relies on the repression of excessive Notch target expression by Su(H)/H complex, then 50% reduction of Su(H) levels should diminish the action of both of these complexes and aggravate the phenotypes of the N/+ and H/+ sensitized backgrounds. Furthermore the increased dosage in Su(H)-GFP heterozygotes should tilt the equilibrium towards more Su(H)/H complexes and improve the H/+ phenotype, instead of intensifying it (Fig. 2H). The authors should make an attempt (in the Discussion) to account for these aberrant behaviors of changing the Su(H) gene dosage, both down and up. A trivial explanation for the Su(H)-GFP effect would be that the GFP fusion has compromised Su(H) function on chromatin, so the fusion acts like the AOS alleles, by sequestering Nicd and H off DNA. Can Su(H)-GFP complement null Su(H) alleles? (Is it functional?)

Some specific items that need to be addressed:

(1) I recommend deleting the word "Notch" from the title. The described mutants do not act via "Notch Cofactor Titration", they act via Su(H) cofactor titration, one of which is Notch.

(2) Avoid the use of phrases like "the presence of a single Su(H) allele" or "Drosophila with a single copy of the AOS-like Su(H) allele" (eg in the abstract and in the second paragraph of the Discussion). They do not clearly convey whether you are talking about the mutant allele heterozygous over the wildtype or over a deletion (the presence of a single mutant allele all on its own). Use the more common genetic terminology "heterozygosity for the Su(H) point mutant allele", which implies over wildtype.

(3) A control is missing from the data of Fig. 7D. What are the levels of Tbp, Hey1 and Hes1 expression in the original MT line before CRISPR knockout of the RBPJ gene?

(4) In Fig. 6F, the bottom panel should read "WB: a-Flag", not "IP: a-Flag". It would also help in this and the next two sub-figures (6G and 6H) to add the word "Input" where appropriate.

(5) In Fig. 1C, please include the mouse mutant names in blue lettering, since they are encountered many times in the ensuing experiments.

(6) With regard to Fig S3: Can cycloheximide block experiments be carried out for so long? Can cells survive without any protein synthesis for 3 days? Do we conclude that the half-life of Rbpj protein is >3 days?

(7) In the Methods section: Is there a reason why the two EMSA experiments (purified proteins vs in vitro synthesized) were performed with a different target oligonucleotide?

(8) The qRT-PCR protocol is written twice in the Methods section. Delete one.

**Have all data underlying the figures and results presented in the manuscript been provided?**

Reviewer #1: Yes

Reviewer #2: Yes

Reviewer #3: Yes

PLOS authors have the option to publish the peer review history of their article (what does this mean?). If published, this will include your full peer review and any attached files.

Reviewer #1: No

Reviewer #2: No

Reviewer #3: No

---

## [Decision Letter · Decision Letter 1]

11 Jul 2022

Dear Dr Kovall,

We are pleased to inform you that your manuscript entitled "A Drosophila Su(H) Model of Adams-Oliver Syndrome Reveals Cofactor Titration as a Mechanism Underlying Developmental Defects" has been editorially accepted for publication in PLOS Genetics. Congratulations!

You'll see (below) that Reviewer #3 has some remaining textual/presentation recommendations. You can attend to these as you prepare your final draft for the production team (the editorial team will not need to re-evaluate).

Yours sincerely,

Pablo Wappner

Associate Editor

PLOS Genetics

Gregory P. Copenhaver

Editor-in-Chief

PLOS Genetics

Comments from the reviewers (if applicable):

Reviewer's Responses to Questions

**Comments to the Authors:**

Reviewer #1: This revised version is now deemed acceptable

Reviewer #3: The authors have made major changes to the manuscript, adding data and improving the presentation. As it stands, it constitutes a coherent body of evidence that the AOS and AOS-like alleles of CSL proteins are dominant-negative (DN) and work via sequestering Notch and negative cofactors away from their target genes. I have a few recommendations for improving the presentation:

(1) By reading the present version, it clarified my earlier bewilderment about why Su(H) AOS-like alleles enhance N/+ and H/+ phenotypes, whereas a Su(H) null suppresses them. The authors still seem to puzzle about it: on p. 30 of the Discussion they state "Currently, we do not have a good molecular explanation for how decreasing the expression levels of the common transcription factor could suppress both Notch- and Hairless-dependent phenotypes". I propose that these opposite effects, both of which are seen when one of the cofactors, H or N, is limiting, stem from a competition between the two cofactors for binding Su(H). To state it more clearly, when one of H, N is limiting (heterozygous), the other (the non-limiting) gains an advantage to bind on target genes "aberrantly". If the dosage of wt Su(H) is simply halved, this advantage is compromised, since less tether is present to mediate this aberrant binding (hence the phenotype is suppressed). If the dosage of wt Su(H) is halved but a DN Su(H) variant is also present, the imbalance between H and N is accentuated, because the limiting cofactor now becomes even more limiting because some of it is titrated away by the DNA-unbound DN Su(H). The non-limiting cofactor now re-gains its advantage to bind to target genes aberrantly and generate a more severe dominant phenotype.

(2) The new experiment in Fig. 7E is also consistent with this model. In the paragraph that describes this experiment (p. 25-26), I recommend changing the phrase:

"Because transfection of increasing amounts of WT Rbpj results in reduced reporter activity (Fig 7E), which has been reported previously35,36 and is likely due to titration of nuclear components," in two ways: (1) cite Fig. 7E AND Suppl. Fig. 5B in the parenthesis and (2) delete the rather vague statement that this squelching is "likely due to the titration of nuclear components". The authors go on to show that this squelching is mostly due to the titration of Nicd, which seems to be limiting in this reporter assay setup (and not due to titration of some general transcription factors).

(3) Also, in the penultimate line of this figure's legend (p. 39) the authors wrongly state: "KRS/NBM have no effect on reporter activity (panel v) due to the inability of this construct to interact with NICD and activate transcription". The real reason that this construct does not squelch the reporter activity is its inability to interact with Nicd and titrate it off the reporter (not "activate transcription").

(4) Another hard-to-follow sentence in the abstract could be improved by changing one word and adding one comma:

"Importantly, genetic studies support a model that heterozygous Drosophila with the AOS-like Su(H) allele behave in an opposing manner as heterozygous flies with a Su(H) null allele due to a dominant activity of sequestering either the Notch co-activator or the antagonistic Hairless co-repressor" change to:

Importantly, genetic studies support a model that heterozygous Drosophila with the AOS-like Su(H) allele behave in an opposing manner to heterozygous flies with a Su(H) null allele, due to a dominant activity of sequestering either the Notch co-activator or the antagonistic Hairless co-repressor.

(5) Finally, in Fig. 4 the various sensory organ primordia shown don't exactly qualify as "macrochetae" (= long bristles). The wing margin bristles are tiny and some of the other sensilla are of the campaniform and chordotonal type (not bristles at all). I would change to "sensory organs" or "sensilla" as a collective term.

**Have all data underlying the figures and results presented in the manuscript been provided?**

Reviewer #1: Yes

Reviewer #3: Yes

PLOS authors have the option to publish the peer review history of their article (what does this mean?). If published, this will include your full peer review and any attached files.

Reviewer #1: No

Reviewer #3: No

**Data Deposition**

http://datadryad.org/submit?journalID=pgenetics&manu=PGENETICS-D-21-01369R1

**Press Queries**

---

## [Editor Report · Acceptance letter]

9 Aug 2022

PGENETICS-D-21-01369R1 

A Drosophila Su(H) Model of Adams-Oliver Syndrome Reveals Cofactor Titration as a Mechanism Underlying Developmental Defects 

Dear Dr Kovall, 

We are pleased to inform you that your manuscript entitled "A Drosophila Su(H) Model of Adams-Oliver Syndrome Reveals Cofactor Titration as a Mechanism Underlying Developmental Defects" has been formally accepted for publication in PLOS Genetics! Your manuscript is now with our production department and you will be notified of the publication date in due course.

With kind regards,

Zsofia Freund

PLOS Genetics

On behalf of:
